# Robust Graph Condensation via Classification Complexity Mitigation

**Jiayi Luo**[1], **Qingyun Sun**[1], **Beining Yang**[2],
**Haonan Yuan**[1], **Xingcheng Fu**[3], **Yanbiao Ma**[4], **Jianxin Li**[1],[*] **Philip S. Yu**[5]

[1]SKLCCSE, School of Computer Science and Engineering, Beihang University
[2]Laboratory for Foundations of Computer Science, University of Edinburgh
[3]Key Lab of Education Blockchain and Intelligent Technology, Guangxi Normal University
[4]Gaoling School of Artificial Intelligence, Renmin University of China
[5]Department of Computer Science, University of Illinois, Chicago
`{luojy,sunqy,yuanhn,lijx}@buaa.edu.cn`
`b.yang-32@sms.ed.ac.uk,fuxc@gxnu.edu.cn,ybma1998@ruc.edu.cn,psyu@uic.edu`

## Abstract

Graph condensation (GC) has gained significant attention for its ability to synthesize smaller yet informative graphs. However, existing studies often overlook the robustness of GC in scenarios where the original graph is corrupted. In such cases, we observe that the performance of GC deteriorates significantly, while existing robust graph learning technologies offer only limited effectiveness. Through both empirical investigation and theoretical analysis, we reveal that GC is inherently an intrinsic-dimension-reducing process, synthesizing a condensed graph with lower classification complexity. Although this property is critical for effective GC performance, it remains highly vulnerable to adversarial perturbations. To tackle this vulnerability and improve GC robustness, we adopt the geometry perspective of graph data manifold and propose a novel **M**anifold-constrained **R**obust **G**raph **C**ondensation framework named **MRGC**. Specifically, we introduce three graph data manifold learning modules that guide the condensed graph to lie within a smooth, low-dimensional manifold with minimal class ambiguity, thereby preserving the classification complexity reduction capability of GC and ensuring robust performance under universal adversarial attacks. Extensive experiments demonstrate the robustness of MRGC across diverse attack scenarios.

## 1 Introduction

Recently, Graph Condensation (GC) [17, 61] has emerged as a promising approach to enhance the training efficiency of Graph Neural Networks (GNNs) by condensing large graphs into smaller ones. These smaller yet highly informative synthesized graphs enable GNNs trained on them to achieve performance comparable to models trained on larger original graphs [20, 41, 54]. This has facilitated the adoption of GC in areas like neural architecture search [33] and graph continual learning [43].

However, the quality of the condensed graph largely depends on the original graph while existing GC methods assume a clean original graph. As shown in Figure 1(a), when the original graph is attacked, the quality of the condensed graph deteriorates, adversely impacting the applications of GC in real-world scenarios where noise and attackers are prevalent [53]. Nevertheless, GC robustness remains largely unexplored. RobGC [16] is the first attempt to tackle this issue. While RobGC effectively addresses structure attacks, its dependence on structure learning and label propagation limits its defense against feature and label attacks [17]. Benchmark [20] also reveals that GC is

---

[*]Corresponding author.

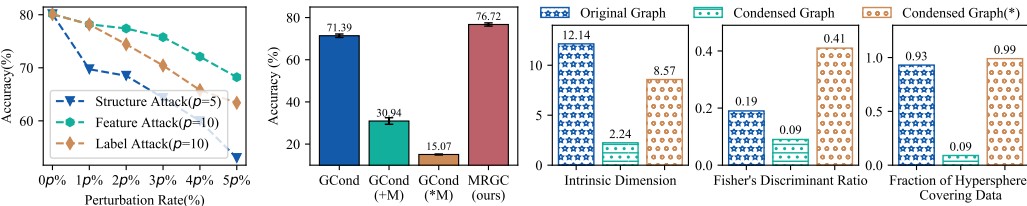

|  (a) GCond under attacks. | (b) GC with robust GNN. | (c) Classification complexity evaluation. |

Figure 1: Examples in Cora [8] dataset (ratio=2.6%): (a) GCond performance under attacks. (b) The robust GNN is adapted for GC under attacks, with (+M) indicating MedianGCN is the GC backbone and (*M) denoting its use in the condensed graph. (c) Classification complexity evaluation, where (*) means the original graph is under attack during GC. Details can be found in Appendix C.

vulnerable to feature attacks. Three key questions about GC robustness remain unsolved: (*Q1*) Can existing robust graph learning techniques improve GC robustness? (*Q2*) What key property of GC is disrupted by attacks, causing the performance degradation, and can it be theoretically understood? (*Q3*) How to design a defense strategy to counter universal attacks in GC?

*Answer to Q1*: Given that most current training-based GC methods utilize GNN as their backbone [20], one intuitive approach to enhance GC robustness is to leverage existing robust GNN technologies. To investigate this possibility, we conduct two toy cases using GCond [33] as a representative GC method. First, we integrate MedianGCN [8] (a classic robust GNN [10, 70]) as the GC backbone. Second, we train MedianGCN on the condensed graph synthesized by the standard GCond. As shown in Fig. 1(b), both strategies fail to work effectively, with even worse performance than the standard GC method. This may be because most existing robust GNNs enhance robustness using attention-like mechanisms [10], which have been shown to perform poorly even in clean GC scenarios [20, 54, 41]. (Results with more robust GNNs are in Appendix C). This result suggests that existing robust GNN techniques may fail to enhance GC robustness, highlighting the need for an innovative solution.

*Answer to Q2*: Since existing GC research primarily focuses on the classification task [17], we examine how attacks affect the classification-related properties of GC. Inspired by the classification complexity theory [25, 44, 36], which investigates classification problems through the geometric properties of classes with the three key factors are intrinsic dimension, boundary complexity, and class ambiguity [44], we explore how the GC process influences the classification complexity of graphs and its behavior under attack through the lens of this theory. To evaluate the classification complexity of graphs during the GC process, we employ three widely used metrics [44]: Intrinsic Dimension, Fisher's Discriminant Ratio, and the Fraction of Hyperspheres Covering Data (Details are in Appendix C). As shown in Figure 1(c), all metrics decrease after GC with an average reduction of 89.25%, indicating that GC will reduce the classification complexity. However, when adversarial attacks occur, we observe an average increase of 547.54% across all metrics in the condensed graph. This reveals an interesting insight: while GC reduces the classification complexity, adversarial perturbations counteract this classification-complexity-reducing property.

*Answer to Q3*: Based on our analysis, designing a defense strategy to preserve the key property of GC under universal attacks to improve robustness presents three challenges: *How to (1) reduce the intrinsic dimension, (2) minimize the complexity of class boundaries, and (3) resolve class ambiguity to mitigate the increasing classification complexity in the condensed graph under attacks?* To address these, we explore GC robustness from the geometric perspective of graph data manifolds and propose a novel **M**anifold-constrained **R**obust **G**raph **C**ondensation framework, named **MRGC**. To maintain a low intrinsic dimension of the condensed graph *(Challenge 1)*, we designed an Intrinsic Dimension Manifold Regularization Module to constrain the condensed graph in a low-dimensional manifold. To reduce the complexity of class boundaries *(Challenge 2)*, we introduce a Curvature-Aware Manifold Smoothing Module to smooth the class manifold in the condensed graph, thereby simplifying the class boundaries. To relieve class ambiguity *(Challenge 3)*, we develop a Class-Wise Manifold Decoupling Module that mitigates potential class bias by minimizing the overlap between class manifolds. Our main contributions are summarized as follows:

- We empirically and theoretically demonstrate that GC inherently reduces the classification complexity of graphs, a property vulnerable to adversarial attacks targeting the original graph and remains unprotected by existing robust graph learning techniques.

- We adopt a geometric perspective of graph data manifold and propose MRGC to enhance GC robustness by protecting its classification complexity reduction property.
- To the best of our knowledge, we present the first study of the robustness of GC under conditions where features, structure, and labels can all be corrupted. Extensive experiments demonstrate the superior robustness of our proposed MRGC.

## 2  Related Work

**Graph Condensation.** GC significantly enhances the training efficiency and scalability of GNNs [33]. Existing GC methods can be categorized into four types [54, 41]: *(1) Gradient Matching:* GCond [33] serves as a representative framework for these methods, optimizing the condensed graph by minimizing gradient discrepancies between GNNs trained on the original and condensed graphs [32, 65, 15]. *(2) Trajectory Matching:* SFGC [74] and GEOM [72] condense graphs by aligning the training trajectories of parameter distributions in expert GNNs. *(3) Distribution Matching:* These methods minimize the distributional difference between the original and the condensed graphs [39, 38]. *(4) Others:* Various methods, such as Kernel Ridge Regression [63, 64] and Computation Tree [24, 23], are also used for GC. However, few studies have explored GC robustness under attack.

**Robust Graph Neural Network.** Lines of studies have been dedicated to enhancing GNN robustness: *(1) Preprocessing:* These methods leverage certain shared properties of real-world graphs to clean perturbed ones before training [12, 58, 30, 45]. *(2) Modeling:* New GNN architectures are proposed to mitigate the impact of attacks dynamically during training [8, 31, 19, 68]. *(3) Training:* They don't modify the GNN architecture but use training strategies like adversarial training [22] or group training [29, 67] to reduce GNN sensitivity.

**Robust Graph Condensation.** With GC recognized as a promising technique [17, 61], RobGC [16] is the first to investigate GC robustness and propose a defense against structure attacks. Benchmark [20] reveals that feature noise poses great threats in GC. However, a comprehensive understanding of GC robustness and a universal defense against structure, feature, and label attacks remains absent.

## 3  Method

**Problem Formulation.** Consider a poisoned graph $\hat{\mathcal{G}}=\{\hat{\mathbf{X}}, \hat{\mathbf{A}}, \hat{\mathbf{Y}}\}$ with $n$ nodes, where $\hat{\mathbf{X}}$ denotes node features, $\hat{\mathbf{A}}$ denotes adjacency matrix, and $\hat{\mathbf{Y}}$ denotes node labels. The goal is to synthesize a compact graph $\mathcal{G}' = \mathbf{X}', \mathbf{A}', \mathbf{Y}'$ with $n' \ll n$ nodes, in a way that is resilient to adversarial attacks, so that GNNs trained on $\mathcal{G}'$ can still perform well on test nodes in the original graph $\hat{\mathcal{G}}$.

$$\min_{\mathcal{G}'} \mathbb{E}_{\mathbf{\Phi} \sim P_{\mathbf{\Phi}}}[\mathcal{L}_{\text{task}}(g_{\mathbf{\Phi}}(\mathcal{G}'), \hat{\mathcal{G}}_{\text{test}})], \tag{1}$$

where $g_{\mathbf{\Phi}}(\mathcal{G}')$ denotes the GNN trained on $\mathcal{G}'$, and $\mathcal{L}_{\text{task}}$ denotes the task-specific loss.

**Framework**. As depicted in Figure 2, we mitigate the increase in classification complexity induced by attacks and enhance GC robustness from three complementary perspectives: ❶ *Reduce the intrinsic dimension (Section 3.1).* We estimate and constrain the intrinsic dimension of the condensed graph during the GC process. ❷ *Minimize the complexity of class boundaries (Section 3.2).* By regularizing the curvature of class manifolds in the condensed graph, we achieve smoother geometric decision boundaries between classes. ❸ *Resolve class ambiguity (Section 3.3).* We minimize the overlapping volume between class manifolds to reduce classification ambiguity and enhance class separability.

### 3.1  Intrinsic Dimension Manifold Regularization

From Figure 1(c), we empirically observe that adversarial attacks will significantly increase the intrinsic dimension of the condensed graph. Here, we first theoretically analyze the relationship between the intrinsic dimension and the graph condensation process and then propose a differentiable method to estimate the intrinsic dimension of the condensed graph. Finally, we impose constraints on the intrinsic dimension throughout the entire condensing process.

Intrinsic dimension is the minimum number of coordinates required to describe the data [2]. Following the widely adopted manifold assumption which states that high-dimensional data lie on a

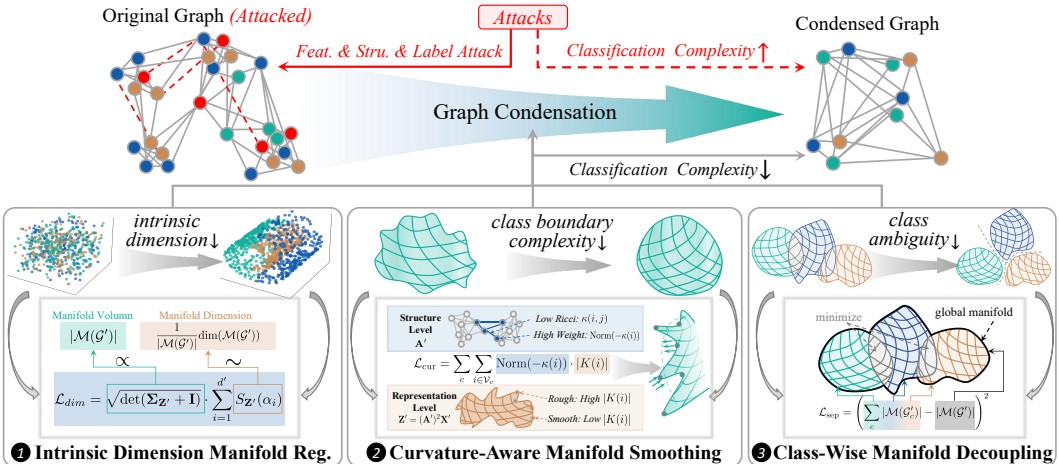

Figure 2: The framework of MRGC, which introduces three complementary graph manifold learning modules into the GC process: constraining the intrinsic dimension, smoothing classification boundaries via manifold curvature limits, and encouraging class manifold decoupling. These modules address the increase in classification complexity within the condensed graph induced by attacks.

lower-dimensional manifold, the intrinsic dimension refers to the dimensionality of this underlying manifold [3, 46]. Here, we begin by presenting the following Theorem 1.

**Theorem 1.** *Given a graph $\mathcal{G}$ with $n$ nodes, let $\mathcal{G}'$ with $n'$ nodes denote the much smaller synthetic graph generated through graph condensation, which is comparable to $\mathcal{G}$ in terms of training GNNs. We have the following:*

$$\mathrm{ID}(\mathcal{G}') < \mathrm{ID}(\mathcal{G}), \tag{2}$$

*where $\mathrm{ID}(\cdot)$ denotes the intrinsic dimension of graph data.*

Theorem 1 indicates that graph condensation is a graph intrinsic dimension decreasing process, synthesizing a condensed graph that lies in a data manifold with a lower dimension. The proof can be found in Appendix A. Building on this, we further propose the following Theorem 2:

**Theorem 2.** *Building on Theorem 1, let $\mathcal{G}'_*$ denotes the synthetic graph generated through graph condensation, where the original graph $\mathcal{G}_*$ is under attack. Then we have:*

$$\mathrm{ID}(\mathcal{G}') < \mathrm{ID}(\mathcal{G}'_*). \tag{3}$$

Theorem 2 indicates that the adversarial attack poses an increasing intrinsic dimension in the condensed graph. Details of the proof are provided in Appendix A.

Through the theoretical analysis in Theorem 1 and Theorem 2, protecting the key intrinsic dimension decreasing characteristic of GC is important for improving its robustness against attacks. In this work, we propose a novel approach to calculate the dimension of the graph manifold where the condensed graph resides and use it as a regularization term to constrain the intrinsic dimension of the condensed graph. Specifically, we use the embedding vectors after two rounds of message passing, $\mathbf{Z}' = (\mathbf{A}')^2 \mathbf{X}'$, as the node representation, which incorporates both feature and structure information. Let $\mathcal{M}(\mathcal{G}')$ denote the manifold on which the condensed graph $\mathcal{G}'$ lies, embedded in $\mathbb{R}^{d'}$. Then, $\{\mathbf{Z}_i\}_{i=1}^{n'}$ represents the discrete set of observations sampled from $\mathcal{M}(\mathcal{G}')$. The intrinsic dimension of $\mathcal{G}'$ is defined as the dimension of $\mathcal{M}(\mathcal{G}')$ under the low-dimension manifold assumption [3], denoted as $\dim(\mathcal{M}(\mathcal{G}'))$. According to [49], $\dim(\mathcal{M}(\mathcal{G}'))$ is given by:

$$\dim(\mathcal{M}(\mathcal{G}')) = \sum_{i=1}^{d'} \int_{\mathcal{M}(\mathcal{G}')} \|\nabla_{\mathcal{M}(\mathcal{G}')} \alpha_i(\mathbf{z}')\| d\mathbf{z}'. \tag{4}$$

Here, $\nabla_{\mathcal{M}(\mathcal{G}')} \alpha_i(\mathbf{z}')$ represents the gradient of coordinate function $\alpha_i$ on $\mathcal{M}(\mathcal{G}')$ at point $\mathbf{z}'$, where $\alpha_i(\mathbf{z}') = z_i'$. Directly solving Eq. (4) requires constructing the explicit function of $\mathcal{M}(\mathcal{G}')$, which is challenging to derive from the discrete observations $\{\mathbf{Z}_i'\}_{i=1}^{n'}$. In this work, we adopt the Laplacian

approximation [69] to solve Eq. (4):

$$\int_{\mathcal{M}(\mathcal{G}')} \|\nabla_{\mathcal{M}(\mathcal{G}')}\alpha_i(\mathbf{z}')\|d\mathbf{z}' \sim |\mathcal{M}(\mathcal{G}')|S_{\mathbf{Z}'}(\alpha_i), \tag{5}$$

where $S_{\mathbf{Z}'}(\alpha_i) = \sum_{p,q} \exp(-\frac{\|\mathbf{Z}'_p - \mathbf{Z}'_q\|_2^2}{2\epsilon^2})(\alpha_i(\mathbf{Z}'_p) - \alpha_i(\mathbf{Z}'_q))$ is the graph Laplacian operator with hyperparameter $\epsilon$, and $|\mathcal{M}(\mathcal{G}')|$ is the volume of the manifold. Based on the geometric interpretation of singular values [1], the volume $|\mathcal{M}(\mathcal{G}')|$ is proportional to the product of the singular values $\{\sigma_i\}_{i=1}^{d'}$ of node representation matrix $\mathbf{Z}'$, i.e., $|\mathcal{M}(\mathcal{G}')| \propto \prod_{i=1}^{d'} \sigma_i$. The covariance matrix of $\mathbf{Z}'$ is defined as $\mathbf{\Sigma}'_{\mathbf{Z}} = \frac{1}{n}(\mathbf{Z}')^\top \mathbf{Z}'$. By the relationship between singular values and eigenvalues, the eigenvalues of $\mathbf{\Sigma}'_{\mathbf{Z}}$ (denoted by $\{\lambda_i\}_{i=1}^{d'}$) are related to the singular values of $\mathbf{Z}$ as $\lambda_i = \frac{\sigma_i^2}{n}$. Thus, the volume $|\mathcal{M}(\mathcal{G}')|$ can be expressed as:

$$|\mathcal{M}(\mathcal{G}')| \propto \sqrt{\det(\mathbf{\Sigma}_{\mathbf{Z}'} + \mathbf{I})}, \tag{6}$$

where $\mathbf{I}$ is to ensure that the covariance matrix $\mathbf{\Sigma}_{\mathbf{Z}'}$ is positive definite, and $\det(\cdot)$ denotes the determinant operation. Finally, the intrinsic dimension manifold regularization loss during graph condensation is defined as:

$$\mathcal{L}_{dim} = \sqrt{\det(\mathbf{\Sigma}_{\mathbf{Z}'} + \mathbf{I})} \cdot \sum_{i=1}^{d'} S_{\mathbf{Z}'}(\alpha_i). \tag{7}$$

### 3.2 Curvature-Aware Manifold Smoothing

The complexity of class boundaries, determined by the geometry of class manifolds, is a key factor in classification difficulty [44]. Attacks can increase this complexity in condensed graphs, weakening GC robustness. To quantify it, we measure the Gaussian curvature [51], where larger absolute values indicate more intricate boundaries. Each node's curvature is computed, and the weighted sum of absolute values is used as a regularization term. Weights reflect each node's influence in message passing and are derived from Ricci curvature [56].

Let $\mathcal{M}(\mathcal{G}'_c)$ denote the class-$c$ manifold containing nodes with label $c$, represented by $\{\mathbf{Z}'_i \mid \mathbf{Y}'_i = c\}$. Our goal is to compute the Gaussian curvature of each node on $\mathcal{M}(\mathcal{G}c')$. We estimate the curvature at node $i$ by fitting a quadratic hypersurface $f_{\mathbf{\Theta}}(\mathbf{o}) = \mathbf{o}^\top \mathbf{\Theta} \mathbf{o}$ to its local neighborhood, where $\mathbf{\Theta}$ is the surface parameter. The curvature is obtained as the determinant of the Hessian of $f_{\mathbf{\Theta}}(\mathbf{\xi})$ [47]. Specifically, we project neighbors of node $i$ onto its tangent space, and use these projections as inputs to fit the hypersurface. The targets are the corresponding projections onto the normal vector at node $i$, capturing how neighbors deviate from the tangent space and thus reflecting local manifold geometry.

We estimate the normal vector $\mathbf{u}_i$ at node $i$ using its $k$-nearest neighbors in Euclidean space [4]:

$$\min_{\mathbf{u}_i} \sum_{j=1}^{k} \left((\mathbf{Z}_i^j - \mathbf{c}_i)^\top \mathbf{u}_i\right)^2, \quad \text{subject to } \mathbf{u}_i^\top \mathbf{u}_i = 1, \tag{8}$$

where $\mathbf{Z}_i^j$ denotes the node representation of $j$-th neighborhood and $\mathbf{c}_i = \frac{1}{k}\sum_{j=1}^{k} \mathbf{Z}_i^j$ is the center of $k$ neighborhoods. To solve Eq. (8), we define the Lagrangian function:

$$\mathcal{L}(\mathbf{u_i}, \lambda) = \sum_{j=1}^{k}((\mathbf{Z}_i^j - \mathbf{c}_i)\mathbf{u}_i)^2 - \lambda(\mathbf{u}_i^\top \mathbf{u}_i - 1), \tag{9}$$

Define $\mathbf{Y} = \mathbf{Z}_i - \mathbf{c}_i \mathbf{1}^\top$. Solving the Karush-Kuhn-Tucker conditions for Eq. (9) shows that the normal vector $\mathbf{u}_i$ corresponds to the eigenvector associated with the smallest eigenvalue of $\mathbf{Y}^\top \mathbf{Y}$; details are provided in Appendix A. Let $\{\lambda_1, \ldots, \lambda_{d'}\}$ and $\{\mathbf{\xi}_1, \ldots, \mathbf{\xi}_{d'}\}$ be the eigenvalues and corresponding eigenvectors of $\mathbf{Y}^\top \mathbf{Y}$, sorted in descending order. Since $\mathbf{Y}^\top \mathbf{Y}$ is symmetric and positive semidefinite, we have $\lambda_1 \geq \cdots \geq \lambda_{d'} \geq 0$ and $\mathbf{\xi}_a^\top \mathbf{\xi}_b = 0$ for all $a \neq b$. The $(d'-1)$-dimensional tangent space at node $i$ is spanned by $\langle \mathbf{\xi}_1, \ldots, \mathbf{\xi}_{d'-1}\rangle$, and the projections of $i$'s $k$ neighbors onto this space form the matrix $\mathbf{O}_i \in \mathbb{R}^{k \times (d'-1)}$, defined as:

$$\mathbf{O}_i = [\mathbf{o}_1, \mathbf{o}_2, \ldots, \mathbf{o}_k]^\top, \tag{10}$$

where each $\mathbf{o}_j \in \mathbb{R}^{d'-1}$ corresponds to the projection of $(\mathbf{Z}_i^j - \mathbf{c}_i)$ onto the tangent space:

$$\mathbf{o}_j = [(\mathbf{Z}_i^j - \mathbf{c}_i) \cdot \boldsymbol{\xi}_1, \dots, (\mathbf{Z}_i^j - \mathbf{c}_i) \cdot \boldsymbol{\xi}_{d'-1}]. \tag{11}$$

Then we propose the following proposition:

**Proposition 1.** *By fitting the quadratic hypersurface $f_\Theta(\mathbf{o})$ via $\min_\Theta \sum_{j=1}^k (\frac{1}{2}\mathbf{o}_j^\top \Theta \mathbf{o}_j - t_j)^2$, where $t_j = (\mathbf{Z}_i^j - \mathbf{Z}_i) \cdot \mathbf{u}_i$ represents the projection along the normal vector, the Gaussian curvature of the class manifold at node $i$ is given by $K(i) = 2\det(\mathrm{Mat}(\mathbf{Q}^{-1}\mathbf{p}))$. Here $\mathbf{Q} \in \mathbb{R}^{(d'-1)^2 \times (d'-1)^2}$ is a fourth-order tensor expressed as a matrix with entries $\mathbf{Q}_{a,b,c,d} = \sum_{j=1}^k o_{ja}o_{jb}o_{jc}o_{jd}$, and $\mathbf{p} \in \mathbb{R}^{(d'-1)^2}$ is a second-order tensor with entries $\mathbf{p}_{a,b} = \sum_{j=1}^k t_j o_{ja}o_{jb}$. The operator $\mathrm{Mat}(\cdot)$ reshapes $\mathbf{Q}^{-1}\mathbf{p}$ into an $(d'-1) \times (d'-1)$ matrix, and $\det(\cdot)$ denotes the determinant operation.*

Proposition 1 provides a closed-form expression for the Gaussian curvature at node $i$, with the proof deferred to Appendix A. However, averaging curvature across all nodes ignores the varying structural roles of individual nodes. In particular, nodes at community boundaries, which serve as bridges for inter-community message passing, have a greater influence on the geometric complexity of class boundaries. To account for this, we reweight each node's Gaussian curvature using Ricci curvature [56], an edge-based metric that reflects structural connectivity. Lower Ricci curvature values on edges indicate stronger bridging roles, highlighting the corresponding node's importance. We adopt the Ollivier definition [48], where the Ricci curvature between nodes $(i,j)$ is defined as $\kappa(i,j) = 1 - \mathcal{W}(m_i^\alpha, m_i^\alpha)/\mathcal{D}(i,j)$, where $\mathcal{W}(\cdot, \cdot)$ is the Wasserstein distance of order 1, $\mathcal{D}(\cdot, \cdot)$ denotes the shortest-path distance, and $m_u^\alpha$ represents the mass distribution, which is defined as:

$$m_i^\alpha(j) = \begin{cases} \alpha, & \text{if } j = i, \\ (1-\alpha)\frac{\mathbf{A}_{ij}}{\deg(i)}, & \text{if } j \in \mathcal{N}(i), \\ 0, & \text{otherwise,} \end{cases} \tag{12}$$

where $\mathcal{N}(i)$ denotes the neighbors of node $i$, $\deg(i) = \sum_{j \in \mathcal{N}(i)} \mathbf{A}_{ij}$, and $\alpha$ is the smoothing parameter and is typically set to $0.5$. The strategy for measuring the Ricci curvature of node $i$ is to average the curvatures of its connected edges [9], expressed as $\kappa(i) = \frac{\mathbf{A}_{ij}}{\deg(i)} \sum_{j \in \mathcal{N}(i)} \kappa(i,j)$. Finally, the Gaussian curvature regularization term, denoted as $\mathcal{L}_{\mathrm{cur}}$, is defined as:

$$\mathcal{L}_{\mathrm{cur}} = \sum_c \sum_{i \in \mathcal{V}_c} \mathrm{Norm}(-\kappa(i)) \cdot |K(i)|, \tag{13}$$

where $\mathrm{Norm}(\cdot)$ denotes min-max normalization to $[0,1]$, and $\mathcal{V}_c$ denotes nodes belonging to class $c$.

### 3.3 Class-Wise Manifold Decoupling

Class ambiguity is the third critical factor that contributes to the classification complexity [25, 44]. To preserve the classification complexity reduction property of GC and improve its robustness, it is essential to avoid class ambiguity in the condensed graph, ensuring that the classes remain well-separated with clear decision boundaries. To avoid the class ambiguity arising in condensed graphs under attacks, we propose measuring the overlap between class manifolds by calculating the difference between the sum of the volumes of individual class manifolds and the volume of the entire data manifold. Minimizing this difference defines our class-wise manifold decoupling objective, which can be expressed as follows:

$$\mathcal{L}_{\mathrm{sep}} = \left( \sum_c |\mathcal{M}(\mathcal{G}_c')| - |\mathcal{M}(\mathcal{G}')| \right)^2, \tag{14}$$

where $|\mathcal{M}(\mathcal{G}_c')|$ represents the volume of class-$c$ manifold. This approach facilitates sufficient decoupling between class manifolds to mitigate class ambiguity, thereby effectively reducing classification complexity in the condensed graph.

**Training Pipeline**. We initialize node features in the condensed graph by randomly selecting non-outlier nodes from the original graph, where outliers are identified based on the Euclidean distances of their features. Other initialization follows [33]. The training loss is as follows:

$$\mathcal{L} = \mathcal{L}_{\mathrm{GC}} + \alpha\mathcal{L}_{\mathrm{dim}} + \beta\mathcal{L}_{\mathrm{cur}} + \gamma\mathcal{L}_{\mathrm{sep}}, \tag{15}$$

where $\mathcal{L}_{\mathrm{GC}}$ denotes the loss function of GC backbone, as MRGC is a plug-and-play framework, and $\alpha, \beta, \gamma$ are hyperparameters. Detailed pipeline are in Appendix D.

**Complexity Analysis.** The Intrinsic Dimension Manifold Regularization Module requires $\mathcal{O}(n'd' + (d')^3)$ operations. The Curvature-Aware Manifold Smoothing consists of two parts: Gaussian curvature computation, with a complexity of $\mathcal{O}(n'((d')^6 + k))$, and Ricci curvature computation, which requires $\mathcal{O}((n')^2)$ operations. The Class-Wise Manifold Decoupling Module has a complexity of $\mathcal{O}(c(d')^3)$, where $c$ denotes the number of classes. **It is worth noting that the practical computational cost remains efficient** because $n'$, the number of nodes in the condensed graph, is small by design. Additionally, we apply PCA [11] to reduce the feature dimensionality before computing the regularization terms, ensuring that $d'$ stays manageable. Details are provided in Appendix D.

## 4 Experiments

### 4.1 Experimental Settings

**Datasets.** We evaluate MRGC [2] and the baselines on five real-world node classification datasets in a transductive setting: Cora [66], CiteSeer [66], PubMed [66], DBLP [6], and Ogbn-arxiv [27]. The data split configuration follows that of [20] for the Cora, CiteSeer, PubMed, and Ogbn-arxiv datasets. For the DBLP dataset, we use the settings from [28], performing random splits with 20 labeled nodes per class for training, 30 per class for validation, and the remaining nodes for testing.

**Attacks.** For the poisoning attacks, we use the widely adopted PRBCD [18, 53] for structure perturbation. For feature perturbation, we randomly select nodes and assign their features by sampling from a normal distribution. For label perturbation, we randomly select a subset of nodes and uniformly flip their labels to other classes. The attack budget is set to $p$ percent of the total number of edges for structure perturbation and $p$ percent of the number of training nodes for feature and label perturbation.

**Baselines.** We evaluate MRGC against various baseline approaches, including five state-of-the-art graph condensation methods: GCond [33], SGDD [65], SFGC [74], GEOM [72] and GCDM [37]. We also compare with RobGC [16], the first robust graph condensation method specifically designed for defending against structure attacks. Furthermore, following [16], we enhance our comparison by incorporating three strong graph denoising techniques as preprocessing steps for GCond: GCond(+J), which removes edges based on Jaccard similarity; GCond(+S), which uses Singular Value Decomposition (SVD) for low-rank approximation to mitigate high-rank noise; and GCond(+K), which integrates $k$-nearest neighbors based on feature similarity into the original graph with $k = 3$.

**Implement Details.** In this experiment, we use GCond [33] as the backbone of MRGC, which is a gradient-matching-based GC method. However, it is important to note that MRGC is compatible with most existing GC methods. The hyperparameters $\alpha$, $\beta$, and $\gamma$ are determined through a grid search from 1e-3 to 1e2 with logarithmic steps of 5. Details can be found in Appendix C and our code. We repeat all the experiments five times and report the average performance and standard deviation. All the experiments are conducted in a single NVIDIA GeForce RTX 3090 24GB GPU.

### 4.2 Robustness Across Varying Condensation Ratios

In this section, we evaluate the impact of different condensation ratios on the robustness of MRGC across the five aforementioned datasets, using three distinct condensation ratios while keeping the attack budgets invariant. The attack budgets for structure, feature, and label attacks are set to 1%, 10%, and 20% for the Cora, CiteSeer, and Ogbn-Arxiv datasets, and 0.1%, 10%, and 20% for the PubMed and DBLP datasets. The results are presented in Table 1.

From the results in Table 1, we have the following three observations: (1) Across all the datasets and condensation ratios except for Ogbn-arxiv at ratios of 0.05% and 0.50%, MRGC achieves the best performance under poisoning attacks. This highlights the robustness of our proposed method in mitigating the negative impact of attacks on the graph condensation process. (2) On the Ogbn-arxiv dataset at condensation ratios of 0.25% and 0.50%, GEOM demonstrates superior performance. This is because trajectory-matching GC methods, such as SFGC and GEOM, achieve significantly

---

[2] Our code is available at `https://github.com/RingBDStack/MRGC`

better results on the clean Ogbn-Arxiv dataset than gradient-matching GC methods [41, 20, 54]. Consequently, despite performance degradation under attack, they retain a relative advantage due to their initially superior performance on the clean dataset. However, SFGC and GEOM exhibit poor robustness on other datasets. (3) Denoising the graph during the preprocessing stage before GC has limited effectiveness, as these methods assume that the node features and labels are clean.

## 4.3 Robustness Across Varying Attack Budgets

To evaluate the robustness of MRGC under varying attack budgets, we fix the condensation ratio to the lowest value for each dataset and adjust the attack budgets for structure, feature, and label attacks independently, while ensuring that the other attack budgets remain consistent with the settings outlined in Section 4.2. The experiments are conducted on the Cora, CiteSeer, PubMed, and DBLP datasets, and the results are shown in Table 2.

As shown in Table 2: (1) MRGC consistently outperforms all baselines across all datasets and attack budget variations. For example, on the CiteSeer dataset, MRGC achieves improvements of approximately 3.98% and 4.75% over the runner-up with label perturbation ratios of 30% and 40%, respectively. This highlights the robustness of MRGC against varying attack intensities. (2) The robustness of MRGC remains stable regardless of the type of attack, effectively defending against structure, feature, and label perturbations. In contrast, RobGC performs well under the structure attacks but shows performance degradation as the intensity of feature and label attacks increases. (3) Gradient matching-based GC methods generally exhibit better robustness compared to trajectory matching and distribution matching GC methods.

## 4.4 Ablation Study

To verify the effectiveness of each component, we compare different ablated versions of MRGC on Cora and CiteSeer datasets with the lowest condensation ratio: (1) MRGC (w/o ID), (2) MRGC (w/o C), and (3) MRGC (w/o D), which respectively remove the Intrinsic Dimension Manifold Regularization, Curvature-Aware Manifold Smoothing, and Class-Wise Manifold Decoupling modules. As the results are shown in Figure 3, all three modules contribute to the performance of MRGC, with the Intrinsic Dimension Manifold Regularization module providing the greatest improvement.

## 4.5 Classification Complexity Study

Based on our analysis in this work, graph condensation acts as a process that reduces classification complexity, whereas attacks disrupt this property, leading to an increase in the classification complexity of the condensed graph. This study aims to verify the ability of our proposed MRGC to mitigate this negative impact. As introduced in Section 1, we measure classification complexity using three

Table 1: Performance comparison under *different condensation ratios* when the training graph is corrupted. S, F, and L represent the attack budgets for structure, feature, and label, respectively. The best results are highlighted in **bold**, while the runner-up results are underlined. OOM denotes out-of-memory (24GB), and OOT denotes out-of-time (24 hours).

| Dataset (S,F,L) | Ratio | GCond | SGDD | SFGC | GEOM | GCDM | GCond (+S) | GCond (+J) | GCond (+K) | RobGC | **MRGC** |
|---|---|---|---|---|---|---|---|---|---|---|---|
| Cora (1,10,20) | 1.30% | 70.69±1.79 | 68.28±0.98 | 39.70±1.61 | 40.43±3.10 | 45.91±4.19 | 68.11±1.27 | 69.29±0.48 | 71.79±0.51 | 71.60±1.64 | **77.43±0.32** |
| | 2.60% | 71.39±0.82 | 68.48±0.62 | 50.10±4.42 | 54.00±4.50 | 49.89±6.01 | 68.82±1.89 | 69.66±1.66 | 72.00±0.75 | 72.19±0.89 | **76.72±0.76** |
| | 5.20% | 71.07±1.42 | 68.63±1.04 | 68.26±0.97 | 70.03±0.40 | 51.37±2.25 | 71.92±2.19 | 70.43±1.79 | 71.75±0.69 | 72.51±1.42 | **74.40±0.29** |
| CiteSeer (1,10,20) | 0.90% | 60.77±2.00 | 47.13±0.75 | 36.23±1.06 | 36.70±1.93 | 46.72±4.49 | 58.49±5.72 | 60.11±1.74 | 58.62±0.37 | 59.88±1.70 | **65.12±0.69** |
| | 1.80% | 61.03±1.28 | 56.12±0.35 | 48.69±1.74 | 47.17±1.54 | 46.01±1.74 | 60.12±3.58 | 59.80±2.68 | 60.58±1.97 | 61.77±2.46 | **63.87±1.04** |
| | 3.60% | 61.08±1.70 | 52.11±0.94 | 62.17±0.38 | 60.67±0.15 | 47.03±1.35 | 61.15±2.73 | 61.88±2.73 | 61.91±0.76 | 62.50±1.65 | **64.44±0.97** |
| PubMed (0.1,10,20) | 0.08% | 70.81±0.74 | 48.75±0.91 | 47.26±2.74 | 50.17±2.52 | 63.81±1.57 | OOM | 70.00±1.01 | 66.48±2.13 | 69.27±0.14 | **74.85±0.34** |
| | 0.15% | 69.02±0.62 | 49.08±0.20 | 64.58±0.79 | 62.90±1.11 | 58.90±4.47 | | 70.31±2.08 | 67.28±1.74 | 71.59±1.62 | **73.11±0.39** |
| | 0.30% | 71.59±1.87 | 51.31±3.14 | 63.18±0.44 | 64.93±1.12 | 61.75±1.09 | | 70.33±2.18 | 68.58±0.42 | 71.71±0.73 | **73.04±0.26** |
| DBLP (0.1,10,20) | 0.11% | 62.50±1.41 | 51.06±3.63 | 41.71±1.27 | 49.71±1.99 | 54.78±7.74 | OOM | 57.13±0.80 | 58.53±1.46 | 62.66±1.46 | **65.11±1.26** |
| | 0.23% | 61.37±1.75 | 44.54±1.77 | 46.95±1.91 | 53.25±1.93 | 54.31±2.36 | | 56.78±1.74 | 60.05±4.44 | 61.65±0.75 | **65.49±1.50** |
| | 0.45% | 62.73±1.20 | 53.45±4.44 | 61.60±0.99 | 60.31±0.66 | 52.87±1.88 | | 57.09±2.40 | 63.58±0.66 | 62.87±1.36 | **64.60±0.18** |
| Ogbn-Arxiv (1,10,20) | 0.05% | 58.16±0.87 | 52.41±1.55 | 55.32±0.51 | 58.38±0.89 | 43.20±2.10 | OOM | 57.15±0.15 | 58.32±0.89 | OOT | **59.38±0.58** |
| | 0.25% | 59.82±0.49 | 57.81±0.77 | 58.37±2.52 | **66.49±0.45** | 52.59±1.47 | | 60.15±1.20 | 60.96±0.42 | | 61.79±0.72 |
| | 0.50% | 60.50±0.31 | 61.04±0.66 | 62.94±1.57 | **67.56±0.22** | 54.91±2.54 | | 59.05±0.43 | 62.12±0.76 | | 63.32±0.34 |

Table 2: Performance under *different attack budgets* targeting the original graph. S, F, and L represent the attack budgets for structure, feature, and label, respectively. The best results are highlighted in **bold**, while the runner-up results are underlined. OOM denotes out-of-memory (24GB).

| Dataset (Ratio%) | Budget | GCond | SGDD | SFGC | GEOM | GCDM | GCond (+S) | GCond (+J) | GCond (+K) | RobGC | MRGC |
|---|---|---|---|---|---|---|---|---|---|---|---|
| Cora (1.30%) | S.5% | 60.59±0.94 | 59.16±1.59 | 52.52±3.39 | 35.03±1.48 | 49.51±2.15 | 60.71±6.21 | 61.90±0.15 | 61.96±2.33 | 62.03±1.80 | **65.57±0.90** |
| | S.10% | 59.09±2.15 | 53.68±2.87 | 37.61±1.32 | 32.97±1.06 | 43.95±3.90 | 60.72±1.44 | 60.96±3.05 | 60.25±4.52 | 61.24±0.68 | **64.02±0.15** |
| | F.20% | 69.36±3.30 | 68.54±1.63 | 37.49±2.43 | 35.80±1.31 | 49.55±2.09 | 66.52±2.36 | 65.23±2.99 | 69.82±3.03 | 71.13±0.81 | **74.79±1.12** |
| | F.30% | 65.22±0.79 | 58.34±1.23 | 37.13±2.42 | 35.60±0.80 | 45.14±4.78 | 57.69±2.36 | 60.06±2.09 | 69.38±5.14 | 69.15±0.42 | **72.36±1.02** |
| | L.30% | 64.97±2.25 | 68.02±0.27 | 32.77±5.40 | 33.57±0.96 | 45.61±6.40 | 67.51±0.44 | 61.59±1.81 | 67.47±2.72 | 65.99±0.87 | **70.47±1.19** |
| | L.40% | 57.49±1.33 | 61.97±2.19 | 28.33±0.93 | 32.63±0.86 | 37.02±1.77 | 57.50±0.40 | 57.29±2.16 | 62.56±1.46 | 57.81±3.69 | **63.89±0.94** |
| CiteSeer (0.90%) | S.5% | 48.51±2.90 | 44.24±2.82 | 35.65±0.78 | 35.30±1.61 | 42.32±1.58 | 49.28±0.80 | 49.21±3.39 | 47.92±2.34 | 48.96±0.78 | **52.91±2.00** |
| | S.10% | 45.06±1.72 | 42.90±1.97 | 33.83±2.14 | 34.67±1.57 | 40.98±4.25 | 42.82±0.74 | 46.69±4.28 | 47.83±4.92 | 47.20±0.60 | **52.72±1.74** |
| | F.20% | 58.26±1.00 | 53.55±3.37 | 39.56±4.34 | 43.50±3.31 | 40.65±4.12 | 53.03±2.73 | 59.32±2.57 | 57.82±4.41 | 61.02±1.51 | **62.76±0.56** |
| | F.30% | 54.45±2.85 | 48.81±4.09 | 37.00±3.87 | 42.20±4.21 | 39.73±2.80 | 55.50±6.73 | 59.49±2.72 | 55.56±2.89 | 56.31±2.50 | **62.41±1.30** |
| | L.30% | 51.69±3.85 | 52.42±3.25 | 45.36±1.89 | 40.93±0.68 | 39.42±3.96 | 51.49±5.81 | 55.35±3.11 | 54.94±5.37 | 49.40±1.41 | **59.33±2.35** |
| | L.40% | 49.25±1.20 | 47.63±1.99 | 31.91±1.25 | 33.03±1.46 | 34.80±0.66 | 51.38±6.36 | 52.31±4.05 | 54.45±2.18 | 48.83±5.45 | **59.20±1.26** |
| PubMed (0.08%) | S.0.5% | 56.54±1.59 | 43.18±2.15 | 54.21±3.43 | 44.47±1.35 | 46.77±3.23 | OOM | 59.02±1.17 | 58.55±1.26 | 57.41±0.62 | **60.57±0.23** |
| | S.1.0% | 55.21±3.89 | 43.55±1.38 | 46.13±2.97 | 44.07±0.40 | 45.38±4.46 | | 58.09±0.36 | 58.28±0.41 | 58.72±1.23 | **58.96±0.78** |
| | F.20% | 71.10±1.01 | 50.85±5.29 | 69.11±3.82 | 44.93±0.23 | 59.48±6.67 | | 68.53±2.26 | 67.78±2.59 | 72.12±2.20 | **73.43±0.79** |
| | F.30% | 67.26±0.74 | 47.72±1.64 | 65.70±0.80 | 60.70±1.13 | 51.20±6.05 | | 63.88±1.92 | 66.18±2.10 | 66.62±1.73 | **68.86±0.34** |
| | L.30% | 68.04±1.83 | 48.62±4.49 | 48.63±1.22 | 49.00±1.74 | 40.70±2.36 | | 71.76±2.26 | 66.83±3.97 | 65.79±2.77 | **73.38±1.58** |
| | L.40% | 56.14±2.64 | 46.22±1.09 | 43.14±3.14 | 45.30±1.90 | 43.72±4.71 | | 58.24±1.09 | 49.96±2.79 | 58.03±1.93 | **58.28±0.31** |
| DBLP (0.11%) | S.0.5% | 62.31±1.45 | 57.28±1.84 | 44.87±5.18 | 46.19±0.66 | 47.75±3.86 | OOM | 55.44±3.06 | 58.10±4.99 | 62.88±2.79 | **65.36±1.81** |
| | S.1.0% | 61.26±0.62 | 58.21±2.92 | 45.13±3.62 | 46.43±0.71 | 43.45±6.30 | | 54.44±2.06 | 58.09±6.10 | 61.89±3.12 | **64.49±0.27** |
| | F.20% | 60.41±2.14 | 62.03±0.53 | 59.69±1.06 | 57.69±0.23 | 49.28±3.30 | | 56.81±1.49 | 60.36±4.63 | 60.39±3.37 | **63.42±1.86** |
| | F.30% | 60.45±1.02 | 55.12±5.03 | 58.70±1.66 | 56.49±4.26 | 45.65±2.55 | | 55.14±1.34 | 61.77±1.49 | 61.51±1.31 | **63.68±0.53** |
| | L.30% | 70.34±1.24 | 64.52±4.15 | 48.84±1.81 | 50.22±0.84 | 53.87±1.30 | | 61.77±1.80 | 69.16±1.10 | 69.39±1.69 | **73.39±0.53** |
| | L.40% | 68.30±1.15 | 63.76±1.29 | 58.63±2.11 | 51.58±2.97 | 37.73±6.47 | | 60.52±1.65 | 50.34±1.85 | 70.66±2.60 | **73.12±0.84** |

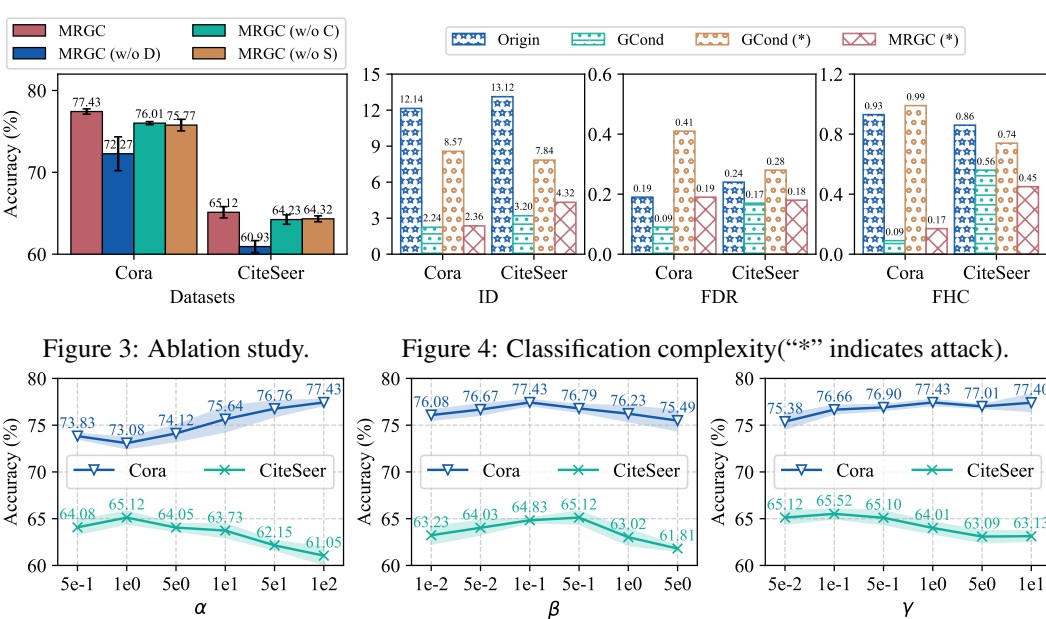

Figure 3: Ablation study.

Figure 4: Classification complexity("*" indicates attack).

Figure 5: Hyperparameters study.

widely adopted metrics: Intrinsic Dimension (ID), Fisher's Discriminant Ratio (FDR), and Fraction of Hyperspheres Covering Data (FHC). Details of these metrics can be found in Appendix C. The experiments are conducted in Cora (2.60%) and CiteSeer (1.80%) and the results are presented in Figure 4. We can see that our proposed MRGC effectively preserves the classification-complexity-reducing property of GC, contributing to the achievement of robust graph condensation.

### 4.6 Hyperparameter Sensitivity Study

In this section, we explore the sensitivity of the hyperparameters $\alpha$, $\beta$, and $\gamma$ for MRGC. In the experiments, we vary the values of $\alpha$, $\beta$, and $\gamma$ on the Cora and CiteSeer datasets with the lowest condensation ratio to examine their impact on model performance. The results are shown in Figure 4.5. As we can see, MRGC performs better when appropriate values are chosen for all hyperparameters, and a wide range of hyperparameters can still yield satisfactory results.

## 5    Conclusion

In this work, we explore GC's robustness against adversarial attacks on features, structures, and labels. Through empirical and theoretical analysis, we discover that GC functions as an intrinsic-dimension-reducing mechanism that creates graphs with lower classification complexity, while this property is susceptible to adversarial attacks. To protect this critical characteristic and improve the robustness of GC, we adopt the geometric perspective of the graph data manifold and propose MRGC, a novel manifold-constrained robust graph condensation framework. Specifically, we introduce three modules that constrain the intrinsic dimension, manifold curvature, and class manifold overlap of the condensed graph, thereby maintaining the classification-complexity-reducing property. Experiments demonstrate MRGC's effectiveness against universal attacks. One limitation is that our focus node classification task, with the graph classification task left for our future work.

## Acknowledgements

The corresponding author is Jianxin Li. The authors of this paper are supported by the National Natural Science Foundation of China through grants No.62225202, and No.62302023. We owe sincere thanks to all authors for their valuable efforts and contributions.

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

# A Proofs and Derivations

## A.1 Proof of Theorem 1

Here, we first restate the theorem:

**Theorem 3.** *Given a graph $\mathcal{G}$ with $n$ nodes, let $\mathcal{G}'$ with $n'$ nodes denote the much smaller synthetic graph generated through graph condensation, which is comparable to $\mathcal{G}$ in terms of training GNNs. We have the following:*

$$\mathrm{ID}(\mathcal{G}') < \mathrm{ID}(\mathcal{G}), \tag{16}$$

*where $\mathrm{ID}(\cdot)$ denotes the intrinsic dimension of graph data.*

*Proof.* We begin by restating the corollary derived from [55], which provides bounds on the probability of successfully learning a class $Y$. Specifically:

**Corollary 1.** *With the training sample $\{x_i \sim X\}_{i=1}^{k}$, the probability of successfully learning the class $X$ is bounded by:*

$$P(g(x) = l_X) \leq 1 - (1 - \frac{1}{2^{ID(X)+1}})^k, \tag{17}$$

*where $k$ represents the number of the training samples, $P(g(x) = l_X)$ denotes the probability of correctly predicting the label of sample $x$ with the trained classifier $g$, and $ID(X)$ is the intrinsic dimension of class $X$.*

These bounds provide insights into the relationship between the intrinsic dimension of the class and the probability of correct classification. Here, we first make the following reasonable assumptions:

**Assumption 1.** *(Graph Condensation Equivalence Assumption). The original and the condensed graphs exhibit equivalent capability in training GNNs [17].*

**Assumption 2.** *(Node Classification Decomposition Assumption). The multi-class classification task for nodes on the graph can be approximated as the union of binary classification tasks for each node type [52].*

**Assumption 3.** *(Optimal Classification Assumption). Under assumption 2, there exists a GNN with optimized parameters that can achieve optimal classification performance for each class on the graph, reaching the upper bound specified in Eq. (17).*

Here, we discuss the probability of classifying the nodes with the label $c$ in the graph $\mathcal{G}$. Under assumption 2 and assumption 3, there exists a GNN classifier $g_{\mathbf{\Phi}}$ with parameter $\mathbf{\Phi}$, having the following:

$$P(g_{\mathbf{\Phi}}(n_c) = l_c) = 1 - (1 - \frac{1}{2^{\mathrm{ID}(\mathcal{G}_c)+1}})^{k_c}, \tag{18}$$

where $\mathcal{G}_c$ denotes the set of nodes belonging to class-$c$ in graph $\mathcal{G}$, $n_c$ represents a test node belonging to class-$c$, $k_c$ is the number of training nodes in $\mathcal{G}_c$, and $\mathrm{ID}(\mathcal{G}_c)$ denotes the intrinsic dimension of $\mathcal{G}_c$. Similarly, for the condensed graph $\mathcal{G}'$, we have the following:

$$P(g_{\mathbf{\Phi}'}(n_c) = l_c) = 1 - (1 - \frac{1}{2^{\mathrm{ID}(\mathcal{G}_c')+1}})^{k_c'}. \tag{19}$$

We can extract the intrinsic dimension $\mathrm{ID}(\mathcal{G}_c)$ and $\mathrm{ID}(\mathcal{G}_c')$ from Eq. (18) and Eq. 19 as follows:

$$\mathrm{ID}(\mathcal{G}_c) = -\left(1 + \log(1 - \sqrt[k_c]{1 - P(g_{\mathbf{\Phi}}(n_c) = l_c)})\right), \tag{20}$$

$$\mathrm{ID}(\mathcal{G}_c') = -\left(1 + \log(1 - \sqrt[k_c']{1 - P(g_{\mathbf{\Phi}'}(n_c) = l_c)})\right). \tag{21}$$

The above expressions establish a mathematical link between the intrinsic dimension of a class of nodes and the probability of successful classification, providing a basis for comparing the original and synthetic graphs.

We subtract Eq. (20) and Eq. (21), yielding:

$$\text{ID}(\mathcal{G}_c) - \text{ID}(\mathcal{G}_c') = -\left(1 + \log(1 - \sqrt[k_c]{1 - P(g_{\Phi}(n_c) = l_c)})\right) \tag{22}$$

$$+ \left(1 + \log(1 - \sqrt[k_c']{1 - P(g_{\Phi'}(n_c) = l_c)})\right) \tag{23}$$

$$= -\log\left((1 - \sqrt[k_c]{1 - P(g_{\Phi}(n_c) = l_c)})\right) \tag{24}$$

$$+ \log\left(1 - \sqrt[k_c']{1 - P(g_{\Phi'}(n_c) = l_c)}\right) \tag{25}$$

$$= \log\left(\frac{1 - \sqrt[k_c']{1 - P(g_{\Phi'}(n_c) = l_c)}}{1 - \sqrt[k_c]{1 - P(g_{\Phi}(n_c) = l_c)}}\right). \tag{26}$$

Based on assumption 1, we have $\Phi = \Phi'$. Additionally, according to the definition of graph condensation, it follows that $k_c' \ll k_c$. Thus we have:

$$\log\left(\frac{1 - \sqrt[k_c']{1 - P(g_{\Phi'}(n_c) = l_c)}}{1 - \sqrt[k_c]{1 - P(g_{\Phi}(n_c) = l_c)}}\right) > 0. \tag{27}$$

This result implies that $\text{ID}(\mathcal{G}_c') < \text{ID}(\mathcal{G}_c)$. Similarly, we can prove that the intrinsic dimension of the nodes in each class decreases after graph condensation. According to [26, 7]: *"a local ID estimator can be used as a global ID estimator by simply averaging over different samples"*. Therefore, when the intrinsic dimension of node sets of each class in the graph are reduced after graph condensation, we obtain $\text{ID}(\mathcal{G}') < \text{ID}(\mathcal{G})$.

This completes the proof. $\qquad\square$

### A.2 Proof of Theorem 2

Here, we first restate the theorem:

**Theorem 4.** *Based on theorem 1, let $\mathcal{G}_*'$ denotes the synthetic graph generated through graph condensation, where the original graph $\mathcal{G}_*$ is under attack. Then we have:*

$$\text{ID}(\mathcal{G}') < \text{ID}(\mathcal{G}_*'). \tag{28}$$

*Proof.* We make the following reasonable assumption:

**Assumption 4.** *Adversarial attacks in the original graph will lead to a decrease in the quality of the condensed graph.*

Based on the proof of theorem 1, for the nodes of class $c$ in the condensed graph, we have the following equation:

$$\text{ID}((\mathcal{G}_*')_c) - \text{ID}(\mathcal{G}_c') = \log\left(\frac{1 - \sqrt[k_c']{1 - P(g_{\Phi'}(n_c) = l_c)}}{1 - \sqrt[(k_c')_*]{1 - P(g_{\Phi_*'}(n_c) = l_c)}}\right). \tag{29}$$

Due to $(k_c')_* = k_c'$ and $P(g_{\Phi_*'}(n_c)) < P(g_{\Phi'}(n_c))$ under assumption 4, we can get:

$$\log\left(\frac{1 - \sqrt[k_c']{1 - P(g_{\Phi'}(n_c) = l_c)}}{1 - \sqrt[(k_c')_*]{1 - P(g_{\Phi_*'}(n_c) = l_c)}}\right) > 0, \tag{30}$$

which indicates $\text{ID}((\mathcal{G}_*')_c) > \text{ID}(\mathcal{G}_c')$. Similar to the proof of theorem 1, we can get $\text{ID}(\mathcal{G}_*') > \text{ID}(\mathcal{G}')$ according to [26, 7].

This completes the proof. $\qquad\square$

## A.3 Solving Eq. (9)

We restate the Eq. (9):

$$\mathcal{L}(\mathbf{u_i}, \lambda) = \sum_{j=1}^{k}((\mathbf{Z}_i^j - \mathbf{c}_i)\mathbf{u}_i)^2 - \lambda(\mathbf{u}_i^\top \mathbf{u}_i - 1), \tag{31}$$

where $\lambda$ is the Lagrange multiplier. Let $\mathbf{Y} = \mathbf{Z}_i - \mathbf{c}_i\mathbf{1}^\top$. Using this substitution, Eq. (9) can be reformulated as:

$$\mathcal{L}(\mathbf{u_i}, \lambda) = \mathbf{u}_i^\top \mathbf{Y}_i^\top \mathbf{Y}\mathbf{u}_i - \lambda(\mathbf{u}_i^\top \mathbf{u}_i - 1). \tag{32}$$

Thus, the original problem can be treated as the following optimization problem:

$$\min_{\mathbf{u}_i} \mathbf{u}_i^\top (\mathbf{Y}^\top \mathbf{Y})\mathbf{u}_i, \quad \text{subject to } \mathbf{u}_i^\top \mathbf{u}_i = 1. \tag{33}$$

Since $\mathbf{Y}^\top \mathbf{Y}$ is evidently a symmetric matrix, the Karush–Kuhn–Tucker (KKT) conditions for this optimization problem are expressed as:

$$\begin{cases} 2\mathbf{Y}^\top \mathbf{Y}\mathbf{u}_i - 2\lambda\mathbf{u}_i = 0 \\ \mathbf{u}_i^\top \mathbf{u}_i - 1 = 0 \end{cases}, \tag{34}$$

indicating that $\mathbf{u}_i$ must satisfy the eigenvalue equation. Therefore, the solution for $\mathbf{u}_i$ corresponds to the eigenvectors of $\mathbf{Y}^\top \mathbf{Y}$, subject to the unit norm constraint $\mathbf{u}_i^\top \mathbf{u}_i = 1$. Consequently, the problem described in Eq. (9) is equivalent to the following:

$$\mathcal{L}(\mathbf{u_i}, \lambda) = \mathbf{u}_i^\top \mathbf{Y}_i^\top \mathbf{Y}\mathbf{u}_i - \lambda(\mathbf{u}_i^\top \mathbf{u}_i - 1) \tag{35}$$

$$= \mathbf{u}_i^\top \lambda\mathbf{u}_i - 0 \tag{36}$$

$$= \lambda \tag{37}$$

This result implies that the solution for $\mathbf{u}_i$ corresponds to the eigenvectors of $\mathbf{Y}^\top \mathbf{Y}$ associated with its smallest eigenvalues, normalized to satisfy the unit norm constraint. This completes the derivation.

## A.4 Proof of Proposition 1

Here, we first restate the proposition:

**Proposition 2.** *By fitting the quadratic hypersurface $f_{\boldsymbol{\theta}}(\mathbf{o})$ in the tangent space via $\min_{\boldsymbol{\theta}} \sum_{j=1}^{k}(\frac{1}{2}\mathbf{o}_j^\top \boldsymbol{\theta}\mathbf{o}_j - t_j)^2$, where $t_j = (\mathbf{z}_i^j - \mathbf{z}_i) \cdot \mathbf{u}_i$ represents the projection along the normal vector $\mathbf{u}_i$, the Gaussian curvature of the manifold $\mathrm{M}((\mathcal{G}'_*)^c)$ at node $i$ is given by $\mathrm{K}_{\mathrm{G}}(i) = \frac{1}{2}\det(\mathrm{Mat}(\mathbf{Q}^{-1}\mathbf{p}))$, where $\mathbf{Q} \in \mathbb{R}^{m^2 \times m^2}$ is a fourth-order tensor expressed as a matrix with entries $\mathbf{Q}_{a,b,c,d} = \sum_{j=1}^{k}\mathbf{o}_{ja}\mathbf{o}_{jb}\mathbf{o}_{jc}\mathbf{o}_{jd}$. The vector $\mathbf{p} \in \mathbb{R}^{m^2 \times 1}$ is a second-order tensor with entries $\mathbf{p}_{a,b} = \sum_{j=1}^{k} t_j\mathbf{o}_{ja}\mathbf{o}_{jb}$. The operator $\mathrm{Mat}(\cdot)$ reshapes $\mathbf{Q}^{-1}\mathbf{p}$ into an $m \times m$ matrix, and $\det(\cdot)$ denotes the determinant.*

*Proof.* Our goal is to determine the parameter $\boldsymbol{\Theta}$ that defines the quadratic hypersurface in the tangent space. The Gaussian curvature of this hypersurface can then be expressed as the determinant of the Hessian matrix associated with the quadratic hypersurface, i.e., $\det(\boldsymbol{\Theta})$ [47]. Our objective function is:

$$E(\boldsymbol{\Theta}) = \min_{\boldsymbol{\theta}} \sum_{j=1}^{k}(\frac{1}{2}\mathbf{o}_j^\top \boldsymbol{\theta}\mathbf{o}_j - t_j)^2. \tag{38}$$

$E(\boldsymbol{\Theta})$ quantifies the discrepancy between the predicted value and the target (*image in a two-dimensional space, the tangent space is a plane, and the target value to be fitted corresponds to the projection of the node onto the normal vector*). To simply Eq. 38, we perform an equivalent transformation to express the objective function in terms of matrix traces:

$$E(\boldsymbol{\Theta}) = E(\hat{\boldsymbol{\Theta}}) = \mathrm{tr}\left[(\frac{1}{2}\mathbf{O}^{(i)}\hat{\boldsymbol{\Theta}} - \mathbf{T})^\top(\frac{1}{2}\mathbf{O}^{(i)}\hat{\boldsymbol{\Theta}} - \mathbf{T})\right], \tag{39}$$

where tr$(\cdot)$ is the trace operator, $\mathbf{T} \in \mathbb{R}^{k \times 1}$ is the target vector:

$$\mathbf{T} = [t_1, t_2, \ldots, t_k], \tag{40}$$

$\mathbf{Q} \in \mathbb{R}^{k \times (d'-1)^2}$ is defined based on the Kronecker product:

$$\mathbf{O}^{(i)} = \begin{bmatrix} \mathbf{o}_{11} \cdot \mathbf{o}_{11}, & \cdots, & \mathbf{o}_{11} \cdot \mathbf{o}_{1d'-1}, & \cdots, & \mathbf{o}_{1d'-1} \cdot \mathbf{o}_{11}, & \cdots, & \mathbf{o}_{1d'-1} \cdot \mathbf{o}_{1d'-1} \\ \mathbf{o}_{21} \cdot \mathbf{o}_{21}, & \cdots, & \mathbf{o}_{21} \cdot \mathbf{o}_{2d'-1}, & \cdots, & \mathbf{o}_{2d'-1} \cdot \mathbf{o}_{21}, & \cdots, & \mathbf{o}_{2d'-1} \cdot \mathbf{o}_{2d'-1} \\ \vdots & \vdots & \vdots & \vdots & \vdots & \vdots & \vdots \\ \mathbf{o}_{k1} \cdot \mathbf{o}_{k1}, & \cdots, & \mathbf{o}_{k1} \cdot \mathbf{o}_{km}, & \cdots, & \mathbf{o}_{kd'-1} \cdot \mathbf{o}_{k1}, & \cdots, & \mathbf{o}_{kd'-1} \cdot \mathbf{o}_{kd'-1} \end{bmatrix}, \tag{41}$$

and $\hat{\boldsymbol{\Theta}} \in \mathbb{R}^{(d'-1)^2}$ represents the vectorized form of the matrix $\boldsymbol{\Theta}$. Specifically, $\hat{\boldsymbol{\Theta}}$ is obtained by concating all rows of $\boldsymbol{\Theta}$ into a single vector, defined as:

$$\hat{\boldsymbol{\Theta}} = [\boldsymbol{\Theta}_1, \boldsymbol{\Theta}_2, \ldots, \boldsymbol{\Theta}_m]^\top \tag{42}$$

By calculating $\nabla_{\hat{\boldsymbol{\Theta}}} E(\hat{\boldsymbol{\Theta}})$, we have:

$$\nabla_{\hat{\boldsymbol{\Theta}}} E(\hat{\boldsymbol{\Theta}}) = \nabla_{\hat{\boldsymbol{\Theta}}} \mathrm{tr} \left[ (\frac{1}{2}\mathbf{O}^{(i)}\hat{\boldsymbol{\Theta}} - \mathbf{T})^\top (\frac{1}{2}\mathbf{O}^{(i)}\hat{\boldsymbol{\Theta}} - \mathbf{T}) \right] \tag{43}$$

$$= \nabla_{\hat{\boldsymbol{\Theta}}} \mathrm{tr} \left[ \frac{1}{4}\hat{\boldsymbol{\Theta}}^\top (\mathbf{O}^{(i)})^\top \mathbf{O}^{(i)}\hat{\boldsymbol{\Theta}} - \frac{1}{2}\hat{\boldsymbol{\Theta}}^\top (\mathbf{O}^{(i)})^\top \mathbf{T} - \frac{1}{2}\mathbf{T}^\top \mathbf{O}^{(i)}\hat{\boldsymbol{\Theta}} + \mathbf{T}^\top \mathbf{T} \right] \tag{44}$$

$$= \frac{1}{4}\nabla_{\hat{\boldsymbol{\Theta}}} \mathrm{tr} \left( \hat{\boldsymbol{\Theta}}^\top (\mathbf{O}^{(i)})^\top \mathbf{O}^{(i)}\hat{\boldsymbol{\Theta}} \right) - \frac{1}{2}\nabla_{\hat{\boldsymbol{\Theta}}} \mathrm{tr}(\hat{\boldsymbol{\Theta}}^\top (\mathbf{O}^{(i)})^\top \mathbf{T}) - \frac{1}{2}\nabla_{\hat{\boldsymbol{\Theta}}} \mathrm{tr}(\mathbf{T}^\top \mathbf{O}^{(i)}\hat{\boldsymbol{\Theta}}) \tag{45}$$

$$= \frac{1}{4}\nabla_{\hat{\boldsymbol{\Theta}}} \mathrm{tr} \left( \hat{\boldsymbol{\Theta}}^\top (\mathbf{O}^{(i)})^\top \mathbf{O}^{(i)}\hat{\boldsymbol{\Theta}} \right) - \nabla_{\hat{\boldsymbol{\Theta}}} \mathrm{tr}(\hat{\boldsymbol{\Theta}}^\top (\mathbf{O}^{(i)})^\top \mathbf{T}) \tag{46}$$

$$= \frac{1}{2}\hat{\boldsymbol{\Theta}}^\top (\mathbf{O}^{(i)})^\top \mathbf{O}^{(i)} - \mathbf{T}^\top \mathbf{O}^{(i)}. \tag{47}$$

By letting $\nabla_{\hat{\boldsymbol{\Theta}}} E(\hat{\boldsymbol{\Theta}}) = \mathbf{0}$, we can get:

$$\frac{1}{2}(\mathbf{O}^{(i)})^\top \mathbf{O}^{(i)}\hat{\boldsymbol{\Theta}} - (\mathbf{O}^{(i)})^\top \mathbf{T} = \mathbf{0}. \tag{48}$$

Solving Eq. (48), we can get the expression of $\hat{\boldsymbol{\Theta}}$:

$$\hat{\boldsymbol{\Theta}} = 2((\mathbf{O}^{(i)})^\top \mathbf{O}^{(i)})^{-1}(\mathbf{O}^{(i)})^\top \mathbf{T}. \tag{49}$$

Thus, we can get the Gaussian curvature of the manifold $\mathrm{M}((\mathcal{G}'_*)^c)$ at node $i$ is given by $\mathrm{K}_{\mathrm{G}}(i) = \det(\boldsymbol{\Theta}) = 2\det(\mathrm{Mat}(\mathbf{Q}^{-1}\mathbf{p}))$, where $\mathbf{Q} \in \mathbb{R}^{m^2 \times m^2}$ is a fourth-order tensor expressed as a matrix with entries $\mathbf{Q}_{a,b,c,d} = \sum_{j=1}^{k} \mathbf{o}_{ja}\mathbf{o}_{jb}\mathbf{o}_{jc}\mathbf{o}_{jd}$. The vector $\mathbf{p} \in \mathbb{R}^{m^2 \times 1}$ is a second-order tensor with entries $\mathbf{p}_{a,b} = \sum_{j=1}^{k} t_j \mathbf{o}_{ja}\mathbf{o}_{jb}$. The operator $\mathrm{Mat}(\cdot)$ reshapes $\mathbf{Q}^{-1}\mathbf{p}$ into an $m \times m$ matrix, and $\det(\cdot)$ denotes the determinant. This completes the proof. $\qquad\square$

# B  Preliminary

To ensure the completeness and self-contained nature of this paper, we provide an overview of preliminaries relevant to our work.

## B.1  Intrinsic Dimension

The concept of intrinsic dimension (ID) was originally introduced by [5], where it is defined as the number of free parameters required by a hypothetical signal generator to produce a close approximation for each signal in a given collection. According to the low-dimensional manifold hypothesis [21, 13], real-world datasets typically reside (approximately) on low-dimensional manifolds embedded within high-dimensional Euclidean spaces. Specifically, let $\mathbf{X} \in \mathbb{R}^{n \times d}$ represent a dataset embedded in a $d$-dimensional Euclidean space. The data points in $\mathbf{X}$ are assumed to be sampled from a manifold $\mathcal{M}$, which is embedded in $\mathbb{R}^d$, where the intrinsic dimension of the manifold satisfies $\dim(\mathcal{M}) < d$. Here $\dim(\mathcal{M})$ refers to the dimension of the manifold $\mathcal{M}$.

## B.2 Gaussian Curvature

Gaussian curvature is a mathematical concept that quantifies how a surface curves at a given point. It is an intrinsic property of the surface, meaning it depends only on the distances measured along the surface and not on how the surface is embedded in higher-dimensional space. The Gaussian curvature $K$ of a smooth surface in three-dimensional space at a point is the product of the principal curvatures [50], $\kappa_1$ and $\kappa_2$, at the given point: $K = \kappa_1 \kappa_2$. If the surface is represented locally as the graph of a function $f$ via the implicit function theorem, the Gaussian curvature at a point $p$ can be computed as the determinant of the Hessian matrix of $f$, which corresponds to the product of the eigenvalues of the Hessian [50, 47]. Positive Gaussian curvature indicates that a surface curves in the same direction in all tangent directions at a given point, forming a dome-like shape. Negative Gaussian curvature, on the other hand, creates a saddle-like shape. The absolute value of the Gaussian curvature measures the degree of curvature at a point, regardless of its sign, indicating how strongly the surface bends. A larger absolute value reflects a sharper curvature, while a smaller value suggests gentler bending.

## B.3 Quadratic Hypersurface Fitting Process Explanation (Section 2)

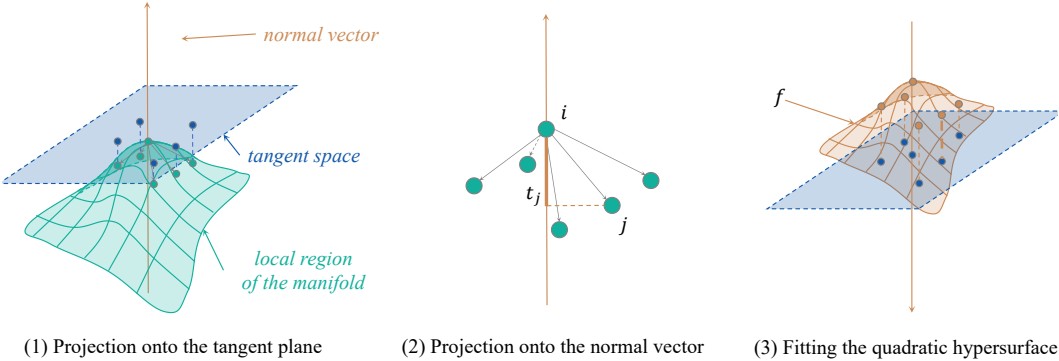

(1) Projection onto the tangent plane     (2) Projection onto the normal vector     (3) Fitting the quadratic hypersurface

Figure 6: Explanation of the Quadratic Hypersurface Fitting Process in Section 3.2

Here, we provide a detailed explanation of the quadratic hypersurface fitting process in Section 3.2. After obtaining the normal vector $\mathbf{u}_i$ and the tangent space $\langle \boldsymbol{\xi}_1, \ldots, \boldsymbol{\xi}_{d'-1} \rangle$. We first project the neighboring points of node $i$ onto its tangent space, as shown on the left side of Figure 6. Next, we fit a quadratic hypersurface using the projected points in the tangent space as input. To fit the hypersurface, we also need to know the output function values. We define the target values as the projections of neighboring nodes onto the normal vector at node $i$. This process is shown in the middle of Figure 6. As illustrated in the figure, the normal vector projections capture the local geometry by quantifying how each neighboring point deviates from the tangent space, ensuring that the fitted hypersurface accurately reflects the manifold's curvature around node $i$. Finally, we fit the quadratic hypersurface by minimizing $\min_{\boldsymbol{\Theta}} \sum_{j=1}^{k} (\frac{1}{2} \mathbf{o}_j^\top \boldsymbol{\Theta} \mathbf{o}_j - t_j)^2$, as introduced in Proposition 1, and this process is illustrated on the right side of Figure 6.

# C Additional Experiment Details and Analysis

## C.1 Details of the Toy Experiments in Section 1

In this section, we detail the toy experiments conducted in section 1.

(1) For the experiments shown in Figure 1(a), we used the Cora dataset with a condensation ratio set to 2.60%. The experiments involved three types of attacks: structure, feature, and label perturbations. For the structure attack, we employed the PRBCD algorithm [18], which introduces structural perturbations to the graph. In the feature attack, we randomly selected nodes from the training set and replaced their features with samples drawn from a normal distribution. For the label attack, we randomly selected a subset of nodes and uniformly flipped their labels to other classes. The attack

budget was set to $p\%$ of the total number of edges for structural perturbations and $p\%$ of the number of training nodes for both feature and label perturbations.

(2) For the experiments presented in Figure 1(b), we also utilized the Cora dataset, maintaining a condensation ratio of 2.60%. Our proposed MRGC employed GCond [33] as the backbone for graph condensation. The attack budgets for the feature, structure, and label perturbations were set to 10%, 1%, and 20% of the corresponding components, respectively.

(3) For the experiments presented in Figure 1(c), we utilized the same dataset and attack budget settings as those used in the experiments shown in Figure 1(b). Here, we provide a detailed explanation of the three metrics [44] used in our analysis:

- Intrinsic Dimension(ID): We employ the MLE [34] intrinsic dimension (ID) estimator for our analysis. Specifically, for a given graph $\mathcal{G} = \{\mathbf{X}, \mathbf{A}, \mathbf{Y}\}$ with $n$ nodes, we use the node embeddings derived after two rounds of message passing as the node representations. These embeddings capture both structural and feature information and are computed as $\mathbf{Z} = \mathbf{A}^2\mathbf{X}$. The resulting set $\mathcal{Z} = \{\mathbf{Z}_i\}_{i=1}^n$ represents the node representations of the graph. The intrinsic dimension of $\mathcal{Z}$, which reflects the graph's intrinsic dimension, is then estimated using the MLE approach as follows:

$$\widehat{\mathrm{ID}(\mathcal{Z})} = -\frac{1}{n}\sum_{z \in \mathcal{Z}}\left(\frac{1}{k}\sum_{i=1}^{k}\log\frac{r_i(z)}{r_k(z)}\right)^{-1}, \tag{50}$$

where $k$ is a hyperparameter that determines the number of nearest neighbors of $z$ considered in the calculation, and we set $k = 8$. The term $r_i(z)$ represents the distance between $z$ and its $i$-th nearest neighbor, while $r_k(z)$ denotes the distance between $z$ and its $k$-th nearest neighbor. For this computation, we employ the Euclidean distance metric to measure the proximity between points.

- Fisher's Discriminant Ratio (FDR): It measures the overlap between the values of the features in different classes and is given by:

$$F1 = \frac{1}{1 + \max_{i=1}^{m} r_{f_i}}, \tag{51}$$

where $r_{f_i}$ is the discriminant ratio for each feature $f_i$. Originally, FDR takes the highest value of $r_{f_i}$, meaning that at least one feature should separate the classes. This paper uses the inverse of this original formula, making the FDR values fall between $(0, 1)$, with higher values representing more complex problems where no single feature can discriminate the classes. An common formula for $r_{f_i}$ in classification task is:

$$r_{f_i} = \frac{\sum_{j=1}^{n_c} n_{c_j}\left(\mu_{f_i}^{c_j} - \mu_{f_i}\right)^2}{\sum_{j=1}^{n_c}\sum_{i=1}^{n_{c_j}}\left(x_i^j - \mu_{f_i}^{c_j}\right)^2}, \tag{52}$$

where $n_{c_j}$ is the number of examples in class $c_j$, and $x_i^j$ is the individual feature value for an example from class $c_j$. $p_{c_j}$ represents the proportion of examples in class $c_j$, and $\mu_{f_i}^{c_j}$ is the mean of feature $f_i$ for class $c_j$, with $\sigma_{f_i}^{c_j}$ as the standard deviation. In our approach, similar to the estimation of intrinsic dimension, we consider the graph $\mathcal{G}$ as a set $\mathcal{Z}$.

- Fraction of Hyperspheres Covering Data (FHC): A topological measure used to assess the coverage of a dataset by hyperspheres. This method builds hyperspheres centered at each data point and progressively increases their radius until they intersect with a data point from a different class. Smaller hyperspheres that are contained within larger ones are eliminated, and FHC is defined as the ratio between the number of remaining hyperspheres and the total number of examples in the dataset. The formula for FHC is given by:

$$\mathrm{FHC} = \frac{\#\mathrm{Hyperspheres}(T)}{n}, \tag{53}$$

where $\#\mathrm{Hyperspheres}(T)$ represents the number of hyperspheres needed to cover the dataset, and $n$ is the total number of examples in the dataset. In this work, we follow the calculation method from Lorena et al. (2019). To determine the radius of a hypersphere for a given data point $x_i$, we first calculate the distance matrix between all examples in the dataset. Specifically, the "nearest

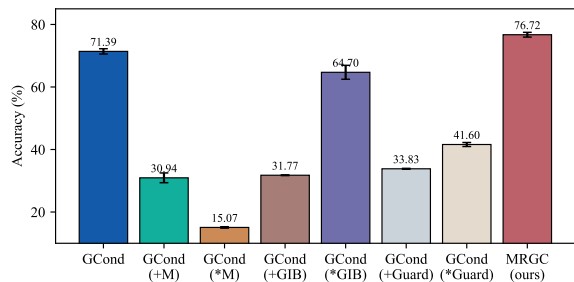

Figure 7: Additional experiments on robust GNNs as GC backbones. Here, (+) indicates that the robust GNNs serve as the backbone of GCond, while (*) signifies that the robust GNN is applied to the condensed graph, which is obtained using the standard GNNs. "M" denotes the MedianGCN [8] and "Guard" represents the GNNGuard [71] .

enemy" of a data point $x_i$ refers to the closest point from a different class. The radius $r_i$ of the hypersphere centered at $x_i$ is then calculated as half of the distance to this nearest point from another class. To compute the radius for $x_i$, we first measure the distance $d_i$ from $x_i$ to another data point $x_j$, then identify the nearest point from a different class for both $x_i$ and $x_j$. If $x_i$ is the nearest point from a different class to $x_j$, the radius is set to half the distance $d_i$. Otherwise, the radius is adjusted based on the distance between the nearest points of the two examples. Once the radii of all hyperspheres are computed, a post-processing step is used to identify which hyperspheres can be absorbed: those completely contained within larger hyperspheres. This ensures that only the most relevant hyperspheres are considered in the final FHC measure. For the distance computation, we continue to use $\mathcal{Z}$, consistent with the approach used in the previous two metrics.

## C.2   Additional Experiments on Robust GNNs as GC Backbones

To further investigate whether existing robust GNN technologies can enhance the robustness of graph condensation, we adopt more robust GNNs as the backbones of GCond. Specifically, we select GNNGuard [71] and GIB [60] as the backbone models for GCond. The results are shown in Figure 7.

As shown in Figure 7, GNNGuard and GIB do not enhance the robustness of GC. Moreover, as mentioned in the Introduction, MedianGCN also fails to improve robustness. This suggests that current robust GNN techniques may not effectively address the robustness of GC, indicating the need for novel solutions to tackle this issue.

## C.3   Dataset Details

In our experiments, we use five real-world datasets: (1) Cora: Contains 2,708 publications classified into seven classes, with 5,429 citation links. Each publication is represented by a binary word vector indicating the presence or absence of 1,433 dictionary words. (2) CiteSeer: Includes 3,312 publications classified into six classes, connected by 4,732 citation links. Each publication is described by a binary word vector over a dictionary of 3,703 words. (3) PubMed: Comprises 19,717 diabetes-related publications classified into three classes, with 44,338 citation links. Each publication is represented by a TF/IDF-weighted word vector from a dictionary of 500 words. (4) DBLP: Contains 17,716 papers, 105,734 citation links, and four labels. (5) Ogbn-arxiv: A directed citation network of computer science papers on ArXiv.

The train/val/test split for the Cora, CiteSeer, PubMed, and Ogbn-arxiv datasets follows the same configuration as the benchmark in [20]. For the DBLP dataset, we use the settings from [28], performing random splits with 20 labeled nodes per class for training, 30 per class for validation, and the remaining nodes for testing.

In our experiments, we use the transactive settings. The condensation ratios for the datasets are configured as follows: For Cora, the ratios are set to 1.30%, 2.60%, and 5.20%. For CiteSeer, they are 0.90%, 1.80%, and 3.60%. In the case of PubMed, the ratios are 0.08%, 0.15%, and 0.30%. Similarly,

for DBLP, the values are 0.11%, 0.23%, and 0.45%. Lastly, for Ogbn-arxiv, the condensation ratios are set to 0.05%, 0.25%, and 0.50%. The condensation ratio refers to the proportion of the number of nodes in the condensed graph to the total number of nodes in the training graph. The statistical information of the datasets is presented in Table 3.

Table 3: The statistics information of datasets.

| Dataset | #Nodes | #Edges | #Classes | #Features | Train/Val/Test |
|---|---|---|---|---|---|
| Cora | 2,708 | 5429 | 7 | 1,433 | 140/500/1,000 |
| CiteSeer | 3,327 | 4732 | 6 | 3,703 | 120/500/1,000 |
| PubMed | 19,717 | 88,648 | 3 | 500 | 60/500/1,000 |
| DBLP | 17,716 | 105,734 | 4 | 1,639 | 80/120/17,516 |
| Ogbn-arxiv | 169,343 | 1,166,243 | 40 | 128 | 90,941/29,799/48,603 |

## C.4 Baseline Details

To better evaluate the robustness of our proposed MRGC, we adopt five types of baselines:

(1) Gradient-matching-based methods.

- GCond [33]: The first gradient matching method that aligns model gradients derived from the training and condensed graphs, ensuring the condensed graph effectively preserves the knowledge for GNN training.

- SGDD [65]: It addresses the issue where GC methods often overlook the impact of structural information from the original graphs. Empirically and theoretically, SGDD demonstrates that synthetic graphs generated by its method are expected to have smaller LED shifts compared to previous works.

(2) Trajectory-matching-based methods.

- SFGC [74]: A structure-free method that aligns model learning behaviors by aligning the training trajectories of parameter distributions in expert GNNs.

- GEOM [72]: It employs a curriculum learning strategy to train expert trajectories with more diverse supervision signals derived from the original graph.

(3) Distribution-matching-based methods.

- GCDM [37]: It minimizes the discrepancy between node distributions of the training and condensed graphs in feature space.

(4) Distribution-matching-based methods. To better highlight the robustness improvement achieved by MRGC, we adopt three state-of-the-art graph denoising methods before applying GC the same as [16]. We choose GCond as the GC backbone.

- GCond(+S) [31]: This method decomposes the noisy graph through Singular Value Decomposition (SVD) and applies low-rank approximation to denoise the graph, reducing the influence of potential high-rank noise.

- GCond(+J) [31]: This method computes the Jaccard similarity for every pair of connected nodes and discards edges whose similarity is below a specific threshold. In our experiments, we set the threshold to 0.01.

- GCond(+K) [30]: It identifies the k-nearest neighbors for each node based on their features and incorporates potentially effective edges into the original graph. In our experiments, we set $k = 3$

(5) Robust Graph Condensation method.

- RobGC [16]: The first proposed robust graph condensation method. It is a plug-and-play defense approach that integrates the structure learning process into the GC process, using the condensed graph as a supervised signal to optimize the original graph structure.

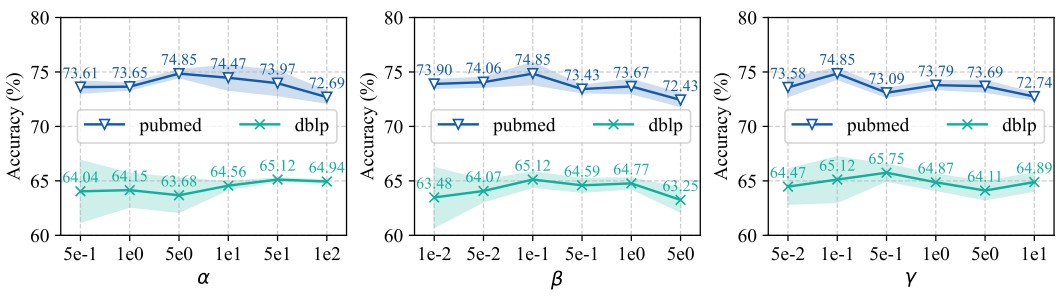

Figure 8: Additional Hyperparameters study.

## C.5 Implement Details

Our proposed MRGC is a plug-and-play graph condensation method, with GCond [33] selected as the GC backbone in our experiments. The hyperparameter settings for each dataset are shown in Table 4.

Table 4: Hyperparameters Setting.

| Dataset | Ratio | $\alpha$ | $\beta$ | $\gamma$ | $k$ | lr(feat) | lr(adj) |
|---------|-------|----------|---------|----------|-----|----------|---------|
| Cora | 1.30% | 1e2 | 1e-1 | 1e0 | 3 | 0.01 | 0.0001 |
| | 2.60% | 5e1 | 1e-1 | 1e0 | 3 | 0.01 | 0.0001 |
| | 5.20% | 5e1 | 1e-2 | 1e-1 | 3 | 0.01 | 0.0001 |
| CiteSeer | 0.90% | 1e0 | 5e-1 | 5e-2 | 3 | 0.01 | 0.0001 |
| | 1.80% | 1e0 | 5e-2 | 5e-2 | 5 | 0.01 | 0.0001 |
| | 3.60% | 1e0 | 5e-2 | 5e-2 | 3 | 0.01 | 0.0001 |
| PubMed | 0.08% | 5e0 | 1e-1 | 1e-1 | 3 | 0.0001 | 0.0001 |
| | 0.15% | 5e0 | 1e-3 | 1e-3 | 5 | 0.0001 | 0.0001 |
| | 0.30% | 5e1 | 1e-3 | 1e-2 | 3 | 0.0001 | 0.0001 |
| DBLP | 0.11% | 5e1 | 1e-1 | 1e0 | 3 | 0.01 | 0.0001 |
| | 0.23% | 1e0 | 5e-2 | 1e0 | 3 | 0.01 | 0.0001 |
| | 0.45% | 1e1 | 5e-1 | 1e-1 | 3 | 0.01 | 0.0001 |
| Ogbn-arxiv | 0.01% | 5e-3 | 1e-9 | 1e-4 | 2 | 0.005 | 0.005 |
| | 0.05% | 5e-3 | 1e-9 | 1e-4 | 3 | 0.005 | 0.005 |
| | 0.50% | 5e-3 | 1e-9 | 1e-4 | 3 | 0.005 | 0.005 |

## C.6 Additional Hyperparameter Sensitivity Study

The sensitivity of additional hyperparameters on the PubMed and DBLP datasets, with a condensation ratio of 0.90% and 0.11%, is shown in Figure 8.

As observed in the DBLP and PubMed datasets, when the hyperparameters are set to suitable values, our MRGC achieves its best performance. Moreover, a wide range of hyperparameter settings still results in consistently superior performance.

# D Algorithm

## D.1 Detailed Complexity Analysis

Here we detailed analysis of the time complexity of our proposed MRGC.

- For the Intrinsic Dimension Manifold Regularization Module, solving the Laplacian approximation requires a computational complexity of $\mathcal{O}(n'd')$, where $n'$ denotes the number of nodes in the condensed graph. Evaluating the volume $|\mathcal{M}(\mathcal{G}')|$ involves computing a determinant, which has a complexity of $\mathcal{O}((d')^3)$, corresponding to the $d'$-dimensional covariance matrix. So the total time complexity for this module is $\mathcal{O}(n'd' + (d')^3)$.

- For the Curvature-Aware Manifold Smoothing Module, the process begins by computing the normal vector and tangent space for each point. This requires solving the eigenvalue decomposition of the matrix $\mathbf{Y}^\top \mathbf{Y} \in \mathbb{R}^{d' \times d'}$, where $\mathbf{Y}$ is constructed from the $k$ nearest neighbors. The time complexity for this step is $\mathcal{O}(k + (d')^3)$. Next, we fit a quadratic hypersurface to obtain the

Gaussian curvature, which involves solving the analytic solution in Proposition 1. Specifically, this step requires computing the matrix inversion of a $(d'-1)^2 \times (d'-1)^2$ matrix $\mathbf{Q}$ and the determinant of the matrix $\text{Mat}(\mathbf{Q}^{-1}\mathbf{p}) \in \mathbb{R}^{(d'-1)\times(d'-1)}$. The time complexity for this is $\mathcal{O}((d'-1)^6 + (d'-1)^3)$. This calculation must be performed for each node, resulting in a time complexity of $\mathcal{O}(n'((d'-1)^6 + k + (d')^3 + (d'-1)^3))$. Finally, we compute the Ricci curvature for each node, with a time complexity of $\mathcal{O}((n')^2)$. Combining these, the total time complexity is $\mathcal{O}(n'((d'-1)^6 + k + (d')^3 + (d'-1)^3 + n'))$. After simplification, this aproximates to $\mathcal{O}(n'((d')^6 + k + n'))$.

- For the Class-Wise Manifolds Decoupling Module, we estimate the volume of the manifold for each class as well as the total graph data manifold. The computational complexity of this module is $\mathcal{O}(c(d')^3)$, where $c$ represents the total number of classes in the graph data, and $d'$ is the dimensionality of the feature space.

The total time complexity of our proposed MRGC is given by: $\mathcal{O}(n'd' + (d')^3 + n'((d'-1)^6 + k + n') + c(d')^3)$. After simplification, the total time complexity of MRGC is approximately to $\mathcal{O}(n'(d')^6 + (n')^2)$. *However, due to the definition of GC, the number of nodes in the condensed graph, $n'$, is significantly smaller than in the original graph, making $n'$ a relatively small number. Furthermore, prior to applying our MRGC, we use PCA [11] to reduce the dimensionality of the condensed graph, which ensures that $d'$ is also kept small. As a result, the actual execution complexity of our approach remains efficient in practice despite the theoretical time complexity.*

### D.2 Detailed Training Pipeline

Here, we detail the overall training pipeline of our proposed MRGCThe overall training pipeline of MRGC is outlined in Algorithm 1. The inputs include the attacked training graph $\hat{\mathcal{G}}$, the hyperparameters $\alpha, \beta, \gamma, k$, and additional hyperparameters determined by the chosen graph condensation backbone. Initially, the node features of the condensed graph are initialized by randomly selecting non-outlier nodes from the original graph that belong to the same class. Outliers are identified based on the Euclidean distance of their features. Let the average distance be denoted as $\mu_d$ and the variance as $\sigma_d^2$. Nodes with a distance to other nodes of the same class greater than $\mu_d + 2\sigma_d$ are classified as outliers. The labels and adjacency matrix of the condensed graph are consistent with those used in [33]. We obtain the node representations by the node features after two rounds of message passing like $\mathbf{Z} = (\mathbf{A}')^2\mathbf{X}$. Then we adopt the PCA [11] to reduce the dimensionality of the node representation to get better performance while significantly reducing the excitation time. Next, we compute the Intrinsic Dimension Manifold Regularization term, calculate the Curvature-Aware Manifold Smoothing Module, and process the Class-Wise Manifold Decoupling Module. These steps yield three manifold regularization terms: $\mathcal{L}_{\text{dim}}$, $\mathcal{L}_{\text{cur}}$, and $\mathcal{L}_{\text{sep}}$. It is worth noting that our proposed MRGC can be integrated with any training-based GC methods. Consequently, the $\mathcal{L}_{\text{GC}}$ is defined by the specific GC backbone it employs. Then, the total training loss is: $\mathcal{L} = \mathcal{L}_{\text{GC}} + \alpha\mathcal{L}_{\text{dim}} + \beta\mathcal{L}_{\text{cur}} + \gamma\mathcal{L}_{\text{sep}}$. Subsequently, the condensed graph is optimized by minimizing $\mathcal{L}$.

## E  Detailed Related Work

Here, we provide a detailed discussion of the related work, which we briefly summarized in Section 2.

**Graph Condensation.** Graph condensation [33] was proposed to enhance the training efficiency and scalability of Graph Neural Networks by synthesizing a much smaller yet highly informative graph. The smaller graph retains the ability to train GNNs effectively, achieving performance comparable to that of GNNs trained on the original, much larger graph. The existing graph condensation studies can be roughly categorized into four lines:

- *Gradient-Maching-Based Methods:* The gradient-matching-based methods were introduced in GCond [33], the first graph condensation method. These methods match the gradient information of the same GNN trained on both the original, larger graph and the condensed, smaller graph, with the goal of minimizing the gradient differences at each training step. However, the standard GCond method ignores the structural information linking the original and condensed graphs. To address this issue, SGDD [65] applies optimal transport theory and designs a method to transfer structural information from the original graph to the condensed graph, thereby achieving smaller shifts in Laplacian Energy Distribution (LED).

---

**Algorithm 1** The overall training pipeline of MRGC.

---

**Require:** Attacked training graph $\hat{\mathcal{G}}$, hyperparameters $\alpha, \beta, \gamma, k$.
**Ensure:** Condensed graph $\mathcal{G}'$
  **for** epoch $= 1, \ldots, E$ **do**
    Initialize the node features, adjacency matrix, and labels of $\mathcal{G}'$.
    $\mathcal{L} = 0$
    Obtain the standard graph condensation loss $\mathcal{L}_{GC}$
    $\mathcal{L} \leftarrow \mathcal{L} + \mathcal{L}_{GC}$
    Calculate the node representation $\mathbf{Z}' = (\mathbf{A}')^2 \mathbf{X}'$
    Dimensionality reduction of the node representation using PCA: $\mathbf{Z}' = \text{PCA}(\mathbf{Z}')$
    **for** $c = 1, \ldots, C$ **do**
      */*Intrinsic Dimension Manifold Regularization*/*
      Compute $\mathcal{L}_{dim}$ by Eq. (7)
      */*Curvature-Aware Manifold Smoothing*/*
      Compute $\mathcal{L}_{cur}$ by Eq. (13)
      $\mathcal{L} \leftarrow \mathcal{L} + (\alpha \mathcal{L}_{dim} + \beta \mathcal{L}_{cur})$
    **end for**
    */*Class-Wise Manifold Decoupling*/*
    Compute $\mathcal{L}_{sep}$ by Eq. (14)
    $\mathcal{L} \leftarrow \mathcal{L} + \gamma \mathcal{L}_{sep}$
    Optimize the condensed graph to minimize $\mathcal{L}$
  **end for**

---

    Furthermore, [32] demonstrates both experimentally and theoretically that one-step gradient matching (i.e., not using the inner loop in gradient matching) can still yield excellent results and significantly improve condensation efficiency. Additionally, MSGC [15] also uses pre-defined structures that allow each condensed node to capture distinct neighborhoods, thereby improving the graph condensation process through better graph structure construction.

- *Trajectory-Matching-Based Methods.* Trajectory-Matching-Based Methods train two separate GNN models on the condensed and original graphs, respectively, and minimize the discrepancy in the training trajectories (i.e., the variation in model parameters) between the final points of these two training trajectories. Unlike gradient-matching-based methods, trajectory-matching-based methods are a multi-step matching approach [17] and often yield better results [54, 20, 41]. SFGC [74] is the first trajectory-matching-based method in graph condensation. It proposes aligning the long-term GNN learning behaviors between the original and condensed graphs, achieving promising results even when the condensed graph has no structural information, thus enabling structure-free graph condensation. Furthermore, GEOM [72] recognizes the limitations of supervision in trajectory matching and highlights that challenging nodes primarily cause the performance gap in GNNs trained on condensed graphs. To tackle this issue, GEOM assesses these difficult nodes and employs curriculum learning to adjust the matching window size during training dynamically.

- *Distribution-Matching-Based Methods.* Distribution-matching-based methods aim to minimize the discrepancy in graph statistic distributions between the condensed and original graphs. To be specific [17], GCDM [37], CaT [40], and PUMA [42] calculate the node distributions in the shared feature space for both the original and condensed graphs, and they optimize by minimizing the maximum mean discrepancy between the corresponding class distributions. Besides this, GDEM [39] works by aligning the eigenbasis as well as the node features of both the real and synthetic graphs, which helps in reducing the spectrum bias that is typically caused in the synthetic graph.

- *Others.* There are also many different graph condensation methods. For instance, to reduce the computational burden in the inner optimization of gradient-matching-based methods, GCSNTK [57] integrates the Graph Neural Tangent Kernel within the Kernel Ridge Regression framework as an alternative. Additionally, KiDD [63] leverages Kernel Ridge Regression by eliminating the non-linear activation function.

**Robust Graph Learning.** Although Graph Neural Networks have shown promising results, numerous studies [53, 58, 31] indicate that they are vulnerable to adversarial attacks, where even small, imperceptible

perturbations in the graph can significantly degrade their performance. Fortunately, several lines of research have shown promising results in enhancing the robustness of Graph Neural Networks, which can be broadly categorized into three main approaches:

- *Preprocessing-Based Methods.* Preprocessing-based methods aim to denoise the graph before training or inference. For example, GCN-Jaccard [58] uses the Jaccard similarity between node feature pairs, removing edges from node pairs whose similarity falls below a predefined threshold. GCN-SVD [12] is based on the widely accepted assumption that real-world graphs often exhibit low-rank properties. It leverages Singular Value Decomposition (SVD) to decompose the adjacency matrix and discard the low singular values. GCN-KNN [31] connects each node to its $k$ most similar neighbors, adding an edge between them if one does not already exist.

- *Modeling-Based Methods.* Modeling-based methods defend against adversarial attacks by modifying the model architecture. RGCN [75] uses Gaussian distributions as the hidden representations of nodes and mitigates the effects of adversarial changes by absorbing them into the variance of the Gaussian distribution. Pro-GNN [31], inspired by the well-known properties of low-rank, sparsity, and feature smoothness, uses structure learning to dynamically adjust the graph structure to enhance GNN robustness. GNNGuard [71] learns how to assign higher weights to edges connecting similar nodes while pruning edges between unrelated nodes. HANG [73] advocates for the use of conservative Hamiltonian neural flows in constructing GNN to improve its robustness.

- *Training-Based Methods.* Training-based methods do not alter the model architecture or the graph; instead, they employ specially designed training strategies to make the model resistant to adversarial attacks. [14, 22, 35] apply adversarial training to reduce the sensitivity of Graph Neural Networks to adversarial perturbations. SG-GSR [29] design a novel group-training strategy to address the loss of structural information and the issue of imbalanced node degree distribution.

**Robust Graph Condensation.** Early research work [59] observed that traditional graph reduction methods faced significant difficulties in effectively defending against PGD [62] attacks. As graph condensation has gained recognition as a more effective and promising graph reduction technique [17, 61] and has been seen as a solution to improve GNNs training efficiency, RobGC [16] became the first study to delve into the robustness of GC in the face of adversarial attacks. This work introduced a defense mechanism that was specifically designed to address and counteract structure-based attacks. Furthermore, benchmark [20] demonstrated that feature noise, which has been a major challenge in Graph Neural Network models, presents even greater difficulties when applied to GC. This highlighted the importance of considering multiple sources of noise and their impact on the effectiveness of GC.

