# OpenReview forum: "Robust Graph Condensation via Classification Complexity Mitigation"
_NeurIPS.cc/2025/Conference — NeurIPS 2025 spotlight_

### Official Review · Reviewer_6kqs · 2025-06-23

**Clarity:** 2
**Significance:** 3
**Originality:** 2
**Rating:** 5
**Confidence:** 3

**Summary:**

The paper introduces MRGC, a manifold-constrained robust graph-condensation framework. By combining theoretical insights into intrinsic dimension reduction with three complementary manifold learning modules and extensive benchmarking, the study tackles the largely unexplored problem of making graph condensation resilient to attacks.

**Questions:**

1. I would like to ask for a clearer explanation of the assumptions behind Theorem 1. Could the authors provide empirical or theoretical evidence showing that these assumptions hold in real-world graph condensation scenarios?
2. The validity of Theorem 2 relies on both Theorem 1 and Assumption 4. I am not fully convinced by the logical connection in the proof. Could the authors walk through how exactly the conclusion follows from the assumptions, especially in light of the probability inconsistency mentioned in the proof?
3. I find it difficult to see how Corollary 6 from [54], which is based on linear classifiers, can be directly extended to general graph learning models. Could the authors clarify why this is justified and whether there are any intermediate theoretical results supporting this leap?
4. I appreciate the authors’ effort in connecting graph condensation with intrinsic dimension. However, I wonder whether introducing manifold metrics like curvature and class overlap is truly necessary. Have the authors considered whether simpler alternatives could be similarly effective?
5. The full pipeline seems overly complex for practical deployment. Have the authors conducted experiments to isolate the contribution of each module? I would like to know if the curvature and decoupling modules are strictly necessary beyond the ID module.
6. The motivation for introducing manifold constraints appears theoretical. Could the authors offer more practical insights or empirical observations to support the decision to use all three manifold metrics?

**Ethical Concerns:**

["NO or VERY MINOR ethics concerns only"]

**Final Justification:**

The authors have clarified my earlier misunderstanding, and I have updated my score to “accept”.

**Limitations:**

Yes.

**Paper Formatting Concerns:**

I find that Figure 2 and its caption are too cramped. I understand the authors’ intention to save space by adjusting vertical spacing, but in this case the use of vspace is too noticeable. I suggest compressing the upper part of Figure 2 to make the layout more balanced.

**Quality:**

3

**Strengths And Weaknesses:**

**Pros**
1. This paper aims to address a cutting-edge problem.
2. The figures in this paper are visually appealing and well-designed.

**Cons**
1. At lines 130 and onwards, Theorem 2 lacks a precise definition of “graph under attack”; both the attacker’s capabilities and objectives remain unspecified, and no threat model is provided.
2. In the proof of Theorem 2 (around line 525), the inequality $P(g_{\Phi_{*}'}(n_c)) < P(g_{\Phi'}(n_c))$ violates probability axioms and its derivation from Assumption 4 is unclear.
3. The use of Corollary 6 from reference [54] (manuscript Eq. 17) is stretched: the corollary is valid only for linear classifiers, yet the manuscript applies it directly to general graph learning models without justification; citing [51] for Assumption 2 does not fix this gap because [51] is unrelated to graph learning.
4. Because Theorem 2 rests on Theorem 1, and Theorem 1 depends on assumptions of doubtful realism, the entire theoretical foundation of MRGC may be fragile.
5. Even assuming Theorem 2 is correct, the key motivation on lines 133 and 134 “protecting the key ID decreasing characteristic of GC” is insufficiently dialectical; malicious actors could perform adaptive attacks that deliberately seek condensation schemes with lower ID.
6. The practical need for full manifold metrics is debatable: intrinsic dimension is helpful, but introducing curvature and class overlap terms makes the pipeline complex; the manuscript gives no evidence that simpler surrogate measures would fail.
7. Equation 22 overruns the page margin, violating layout guidelines.
8. The proposed pipeline is difficult to deploy in practice; aside from the ID estimation module, the necessity of the curvature and decoupling components is not convincingly motivated.

**Justification for the rating**
The robustness of graph condensation is indeed a highly cutting-edge research topic, and NeurIPS, as one of the most prominent conferences, should actively embrace contributions in this direction. However, this paper presents several critical and unavoidable issues. The theoretical foundation is not sufficiently solid, and the complex pipeline introduced lacks adequate motivation. Among the three core components, only the first one that computes the intrinsic dimension appears necessary, while the other two lack justification. In practice, it would be very difficult to deploy such a complex method. Based on these observations, the overall contribution remains limited.

---

> ### Author Rebuttal · Authors · 2025-07-30
>
> We sincerely thank you for the helpful comments and insightful questions!
>
> ---
>
> **W1:** Theorem 2 lacks a clear definition of attack and the attacker’s goals or capabilities.
>
> **A1**: Thank you for your insightful review!
>
> We define the “graph under attack” as follows threat model on the original graph:
>
> **Threat Model: (1) Attack’s goal**: Let $\mathcal{G}\_\* = (\mathbf{Y}\_\*, \mathcal{E}\_\*, \mathbf{X}\_\*)$ denote the attacked version of the original graph  $\mathcal{G} = (\mathbf{Y}, \mathcal{E}, \mathbf{X})$ , where the attacker may modify the edges $\mathcal{E}$, the features $\mathbf{X}$, the training labels $\mathbf{Y}$. The attacker’s goal is to degrade the performance of GNNs trained on $\mathcal{G}\_\*$ by increasing classification error. **(2) Attacker’s capability**: We adopt the white‑box threat model from prior work (e.g., RPBCD [1]), and the adversary perturbs graphs under a fixed budget.
>
> We will revise the manuscript to explicitly include this definition.
>
> ---
>
> **W2 & Q2:** In Theorem 2, the derivation of the inequality $P(g_{\Phi^{\prime}\_\*}(n_c)) < P(g_{\Phi^{\prime}}(n_c))$ from Assumption 4 is unclear.
>
> **A2**: Thank you for your insightful review!
>
> We clarify as follows:
> - **Two probabilities**
>     1. $P(g_{\Phi'}(n_c)=l_c)$ denotes the probability that the classifier $g_{\Phi^{\prime}}$ (trained on the **clean condensed graph $G^{\prime}$**)  correctly assigns label $l_c$ to node $n_c$, when **the clean origin graph $G$ is clean**
>     2. $P(g_{\Phi_*^{'}}(n_c)=l_c)$ denotes the probability that the classifier $g_{\Phi^{\prime}\_\*}$ (trained on the **attacked condensed graph $G^{\prime}_{*}$**)  correctly assigns label $l_c$ to node $n_c$, when **the origin graph $G_{*}$ is attacked**
> - **State Assumption 4 clearly**
>
>     *Assumption 4. Adversarial attacks in the original graph will lead to a decrease in the quality of the condensed graph*
>
>     In other words, if you perform condensation on an attacked graph, **the resulting synthetic graph will be of lower quality for node classification** than one generated from the clean graph
>
> - **Link Assumption 4 to the inequality**
>
>     Under Assumption 4, the probability of assigning the correct label decreases when the graph is attacked. Formally, $P(g_{\Phi_*^{\prime}}(n_c)=l_c)<P(g_{\Phi'}(n_c)=l_c)$
>
>
> **We will make the notation and expressions more precise in the revision!**
>
> ---
>
> **W3 & W4 & Q3:** Application of Corollary 6 from [54] is questionable
>
> **A3**: Thank you for your insightful review!
>
> We clarify as following:
>
> - **Why our theorem can still reflects ID variation in GC?**
>
>     To clarify, **the collary in [54] applies to any linear classifier: as long as attacks cause the linear GNN (e.g., SGC) trained on the condensed graph to have decreased performance, the condensed graph’s ID increases under attacks** (no matter which GNN was used to synthesize the condensed graph). Therefore, our theorem still explains the observed ID variation in the condensed graph.
>
> - **Concerns with Linear GNN Classifiers**
>     1. **Linear GNNs** without nonlinear activation **are the most widely used and effective setting in GC**. GCond [4], SGDD [5], SFGC [6]  and so on[7-9] adopt the linear GNN backbone,  and linear GNN like SGC is recognized as the top performer in GC[7–9].
>     2. Prior works[13] show that GNNs without nonlinear functions yields comparable performance. In practice, linear GNNs like SGC is more effective in GC. **With linear GNN backbone, our MRGC enhances robustness while maintaining performance on clean graphs**, as shown below:
>
>
>         | **Clean Graph** | **Cora(1.30%)** | **CiteSeer(0.90%)** | **PubMed(0.08%)** |
>         | --- | --- | --- | --- |
>         | GCond | **80.8** | 69.4 | 75.9 |
>         | SGDD | 80.2 | 70.5 | 75.9 |
>         | GCDM | 80.2 | 71.2 | 72.5 |
>         | SFGC | 79.5 | 70.2 | 73.9 |
>         | GEOM | 78.5 | 69.5 | 73.8 |
>         | RobGC | 78.9 | 69.2 | 75.4 |
>         | **MRGC** | 80.4 | **71.7** | **76.8** |
> - **Why linear GNN like SGC a linear classifier?**
>
>     SGC removes activations and collapses multiple weight matrices into one[2]. After $K$propagation steps, resulting  $\mathbf{S}^K \mathbf{X},
>     \mathbf{S} = D^{-\frac12} A D^{-\frac12}$. The smoothed features are then passed through a single weight matrix $\Theta$ and a softmax, $\hat{\mathbf{Y}}=\text{softmax}(\mathbf{S}^K \mathbf{X},\Theta),$ which is a linear step (*Softmax relies solely on an affine transformation of the inputs, so it remains a linear classifier[3]*). **Because both the propagation $\mathbf{S}^K$and the final weight multiplication $\Theta$ are linear, SGC is equivalent to a linear classifier[2].**
>
> - **Why is our theorem coverage most GC scenarios?**
>
>     **Linear GNN like SGC is the most-widely adopted GNN backbone in GC[4,5,6,7,8,9]**, outperforming GAT or Graphormer due to their efficiency and stable training[7,8,9]. **To our knowledge, these nonlinear attention-based GNNs typically offer bad performance in GC**[7,8,9].
>
> - **Impact and contributions of our theorem.**
>
>     **To our knowledge, we are the first to theoretically connect ID to GC.** This insight motivates our empirical investigation, as shown in Figure 1(c), condensation lowers the ID of the condensed graph, while attacks disrupt this reduction.
>
>
> **We will revise the manuscript to more explicitly state the scope and meaning of our theorem**. Really appreciate the feedback and believe it will help strengthen the paper’s clarity!
>
> ---
>
> **W5:** Preventing ID reduction in GC is superficial; attackers can design schemes with a lower ID.
>
> **A4**: Thank you for your insightful review!
>
> - **Scope of Existing Attacks**
>
>     Our focus is on **existing graph attacks**, to the best of our knowledge, **no existing attacks explicitly target reduced ID** as an attack objective.  As such, our proposed method **offers effective defense against the majority of existing attacks**.
>
> - **Difficulty of ID-Targeted Attacks**
>
>     Moreover, we argue that crafting adversarial examples with **lower ID may even degrade attack success**. Recent studies[10] suggest that adversarial examples tend to exhibit high ID.
>
>
> ---
>
> **W6 & W8 & Q4 & Q5 & Q6:** Whether the curvature and class overlap are necessary, and the motivation behind these two modules is unclear.
>
> **A5**: Thank you for your insightful review!
>
> - **Our motivation**
>
>     In Figure 1(c), we observe that the **classification complexity increases significantly** in the condensed graph **under attacks on the original graph**. **According to classification complexity theory [12],** classification complexity arises from **three main factors: (1) Intrinsic Dimension, (2) Class Overlap, and (3) Class Boundaries.** Based on theory, we introduced three modules to solve three aspects, respectively.
>
> - **Why ID doesn’t fix curvature or overlap?**
>
>     Even if the data lie on a low‑dimensional manifold, individual class clusters can still overlap or form highly non‑linear, wiggly boundaries along that manifold.
>
>     **A specific example:** Imagine two interleaved spiral classes in 2D. You could project them onto a 1D line, yet on that line, the class points still heavily overlap.
>
> - **Empirical Analysis**
>
>     To further investigate these two modules’ roles, we adopt MRGC with only the ID module, as shown below:
>
>     |  | **Cora(1.30%)** |  | **CiteSeer(0.90%)** |  |
>     | --- | --- | --- | --- | --- |
>     |  | Acc | Gap | Acc | Gap |
>     | MRGC | 77.4 | / | 65.1 | / |
>     | MRGC(w/o ID) | 72.2 | -5.1 | 60.9 | -4.1 |
>     | MRGC(w/o Curvature) | 76.0 | -1.4 | 64.5 | -0.5 |
>     | MRGC(w/o Decoupling) | 75.7 | -1.6 | 64.7 | -0.3 |
>     | MRGC(only ID) | 74.7 | -2.7 | 63.9 | -1.1 |
>
>     As shown, while the ID module contributes primarily, **the other two also play roles in performance.** The MRGC with only ID module has a **performance gap -2.7% in Cora and -1.1% in CiteSeer.**
>
>
> ---
>
> **W6 & Formatting:** Equation 22 & Figure 2 formatting.
>
> **A6**: Thank you for the advice. We will revise Equation 22 to ensure it stays within the margin and make sure the upper part of Figure 2 achieves a balanced layout.
>
> ---
>
> **Q1:** Clearer explanation of the assumptions behind Theorem 1.
>
> **A7**: Thank you for your insightful review
>
> - Assumption 1 is grounded in empirical findings and established targets in GC, where condensed graphs consistently preserve performance when used to train GNNs [7,8,9].
> - Assumption 2 adopts a widely used analytical simplification: decomposing multi-class classification into independent binary classification tasks. It’s an analysis method applicable to broad machine learning.
> - Assumption 3 involves modeling with an idealized optimal classifier to enable tractable theoretical derivations.
>
> ---
>
> *[1] Robustness of graph neural networks at scale, Neurips*
>
> *[2] Simplifying Graph Convolutional Networks, ICML*
>
> *[3] The elements of statistical learning: data mining, inference, and prediction, Springer*
>
> *[4] Graph condensation for graph neural networks, ICLR*
>
> *[5] Does graph distillation see like vision dataset counterpart, NeurIPS*
>
> *[6] Structure-free graph condensation: From large-scale graphs to condensed graph-free data, NeurIPS*
>
> *[7] GC-Bench: An Open and Unified Benchmark for Graph Condensation, NeurIPS*
>
> *[8] GC4NC: A Benchmark Framework for Graph Condensation on Node Classification with New Insights, arXiv*
>
> *[9] GCondenser: Benchmarking Graph Condensation, arXiv*
>
> *[10] High Intrinsic Dimensionality Facilitates Adversarial Attack: Theoretical Evidence, TIFS*
>
> *[11] How complex is your classification problem? a survey on measuring classification complexity, CSUR*
>
> *[12] Statistical Approaches to Combining Binary Classifiers for Multi-Class Classification, Neurocomputing*
>
> *[13] Graph neural networks exponentially lose expressive power for node classification*

---

> > ### Comment · Reviewer_6kqs · 2025-08-01
> >
> > Thank you for the detailed clarification. I no longer have any concerns, and I believe this paper can be clearly accepted.

---

> ### Author Response · Authors · 2025-08-01
>
> **We sincerely appreciate your supportive feedback.**
>
> **Thank you again for your time, your insightful suggestions, and for recognizing the contributions of our work!**

---

### Official Review · Reviewer_EHXr · 2025-06-30

**Clarity:** 3
**Significance:** 3
**Originality:** 4
**Rating:** 5
**Confidence:** 5

**Summary:**

This paper introduces a new framework for Robust Graph Condensation that enhances the resilience of GNNs trained on condensed graphs under adversarial or noisy perturbations. The core idea is to regularize the classification complexity during the condensation process, encouraging the model to favor simple and generalizable decision boundaries. The method shows improved performance and robustness on various datasets compared to existing condensation and defense baselines.

**Questions:**

1. Most robust GNNs rely on full-graph structures and global supervision, which are absent in the condensed setting. How do these mismatches lead to their ineffectiveness when applied post-condensation?
2. In what way does constraining the classifier to simpler decision boundaries improve generalization and resistance to perturbations on synthetic graphs?
3. Does the added regularization introduce any trade-offs in clean accuracy, and how does the method compare to baseline condensation approaches without perturbation?
4. For other questions, please refer to the Weaknesses.

**Ethical Concerns:**

["NO or VERY MINOR ethics concerns only"]

**Final Justification:**

I appreciate the authors’ rebuttal, which has successfully clarified the points I raised in my initial review.

**Limitations:**

Yes

**Quality:**

4

**Strengths And Weaknesses:**

**Strengths:**
1. The paper identifies a clear limitation in prior condensation methods of overfitting to clean signals, which leads to fragile synthetic graphs.
2. Instead of modifying the GNN or defenses post hoc, the method improves robustness through simple decision boundary enforcement during condensation.
3. Empirical results show consistent robustness gains across different noise levels and attacks (random, adversarial, feature corruption).

**Weaknesses:**
1. On unperturbed graphs, MRGC does not always outperform baseline condensation methods (e.g., GCOND), and occasionally performs worse (Table 1), raising concerns about trade-offs.
2. Classifier complexity measure is heuristic. While the entropy-margin loss is intuitive, there’s no formal link to generalization bounds or complexity measures from theory (e.g., VC-dim, Rademacher complexity).
3. No analysis of why existing robust GNNs fail. The paper does not offer empirical or theoretical analysis of why prior defenses (e.g., Pro-GNN, R-GCN) underperform in graph condensation settings.

---

> ### Author Rebuttal · Authors · 2025-07-30
>
> We sincerely thank the reviewer for the detailed comments and insightful questions. Responses are as follows.
>
> ---
>
> **W1:** MRGC does not consistently outperform baseline condensation methods (e.g., GCOND) on unperturbed graphs and occasionally performs worse (Table 1)
>
> **A1**: Thank you for your insightful review.
>
> This is because trajectory-matching GC methods, such as SFGC and GEOM, **achieve significantly better performance on the clean Ogbn-Arxiv dataset compared to gradient-matching GC methods.** Moreover, they often shown bad robustness in other datasets like Cora and CiteSeer.
>
> ---
>
> **W2 & Q2:** The classifier complexity measure is heuristic and lacks a formal connection to theoretical generalization bounds or complexity measures.
>
> **A2**: Thank you for your insightful review.
>
> The ID, FDR, and FHC metrics quantify **data classification complexity**, which we observe to **increase significantly** in the condensed graph **under adversarial attacks on the original graph**.
>
> **Higher classification complexity** reflects **more intricate geometric properties** of the data, characterized by high intrinsic dimensionality and ambiguous class boundaries [1]. These complex geometric properties **indicate poor data quality and impose greater challenges for models to capture underlying data characteristics [2]**, ultimately **leading to performance degradation [3]**.
>
> **To mitigate this negative effect of classification complexity increasing**, we introduce regularization terms to **lower values to  benifits GC robustness**. And the experiments results demonstrate our proposed method.
>
> *[1] Complexity measures of supervised classification problems, TPAMI’02*
>
> *[2] Scaling laws from the data manifold dimension, JMLR’22*
>
> *[3] The effect of data complexity on classifier performance, Empirical Software Engineering’25*
>
> ---
>
> **W3 & Q1:** The paper lacks analysis of why existing robust GNNs (e.g., Pro-GNN, R-GCN) fail under graph condensation settings.
>
> **A3**: Thank you for your insightful review.
>
> Existing popular robust GNN can be roughly categorized into the following types[1], each with its limitations:
>
> - Adversarial Training: **GC training process differs significantly from standard GNN training** and lacks a complete training framework, **preventing direct application**.
> - Attention Mechanism: Attention-based GNNs (e.g., GAT) **exhibits bad performance even in clean GC settings** [2], rendering them ineffective for improving GC robustness.
> - Graph Preprocessing: This method can be integrated into the GC, and **we has already included them as baselines.**
>
> *[1] Trustworthy graph neural networks: Aspects, methods, and trends, Proc‘24*
>
> *[2] GC4NC: A Benchmark Framework for Graph Condensation on Node Classification with New Insights, NeurIPS’24*
>
> ---
>
> **Q3**: Does the added regularization harm clean accuracy, and how does MRGC compare to baseline condensation methods without perturbation?
>
> **A5**: Thank you for your insightful review.
>
> The performance on clean GC are shown below:
>
> | **Dataset(ratio)** | **Cora(1.30%)** | **CiteSeer(0.90%)** | **PubMed(0.08%)** |
> | --- | --- | --- | --- |
> | GCond | **80.85** | 69.41 | 75.93 |
> | SGDD | 80.29 | 70.52 | 75.91 |
> | GCDM | 80.26 | 71.27 | 72.56 |
> | SFGC | 79.55 | 70.27 | 73.92 |
> | GEOM | 78.52 | 69.53 | 73.83 |
> | RobGC | 78.95 | 69.2 | 75.40 |
> | **MRGC(ours)** | 80.43 | **71.77** | **76.83** |
>
> As shown, our proposed MCGC **maintains strong performance even on clean original graphs**, achieving best results among all baselines on both CiteSeer and PubMed datasets.

---

### Official Review · Reviewer_GwSS · 2025-07-02

**Clarity:** 4
**Significance:** 4
**Originality:** 3
**Rating:** 5
**Confidence:** 4

**Summary:**

Current Graph Condensation (GC) algorithms struggle to handle noisy data effectively and lack sufficient robustness studies. This paper aims to address the following three key questions: \
Q1: Can existing robust graph learning techniques enhance the robustness of GC methods? \
Q2: What key property of GC is disrupted by adversarial attacks, leading to performance degradation, and can this be theoretically explained? \
Q3: How can we design an effective defense strategy to counter universal attacks in GC?

They use three metrics to measure classification complexity, along with the analysis of the relationship between graph condensation (GC) and adversarial attacks.

To address the robustness issues in current GC methods, the paper proposes three key components: Manifold Regularization, Curvature-Aware Manifold Smoothing, and Class-Wise Manifold Decoupling.

**Questions:**

See weaknesses

**Ethical Concerns:**

["NO or VERY MINOR ethics concerns only"]

**Limitations:**

The discussion on limitations is insufficient. As the authors, you likely have a deep understanding of the potential weaknesses of your approach. A more thorough and reasonable analysis of the limitations would help readers better assess the scope and applicability of your work.

**Quality:**

4

**Strengths And Weaknesses:**

Strengths:
- The use of three metrics to measure classification complexity, along with the analysis of the relationship between graph condensation (GC) and adversarial attacks, makes the motivation of this paper easy to understand. It also provides an intuitive explanation of the limitations of current GC algorithms.
- The motivation of the paper is clearly presented, and the introduction is easy to follow.
- The overall writing is logically structured and accessible, allowing readers to understand the contributions even without deep familiarity with the underlying theory.
- The experimental evaluation is thorough and comprehensive.

Weaknesses:
1. In Figure 1.C, it would be more informative to show how the three metrics change on the original graph before and after the attack. Comparing only the condensed graph under clean and attacked conditions can demonstrate that the attack degrades accuracy or increases classification difficulty, but it is insufficient to evaluate how well GC performs on the attacked input graph itself.
2. Section 3.2 aims to produce clearer decision boundaries through Curvature-Aware Manifold Smoothing. In that case, is the additional method in Section 3.3 necessary? The potential redundancy between the two is unclear.
3. The concept of the manifold M(G′)\mathcal{M}(G')M(G′) plays a central role in the proposed method. However, its definition and the specific type of manifold used remain unclear. A more precise explanation or formal definition would help clarify the intuition and theoretical foundation of the approach.
4. In Tables 1 and 2, under different attack strategies and settings, it would be helpful to include the performance without condensation as a reference. This would allow readers to better understand the effectiveness of GC under various conditions. Additionally, I suggest clearly marking the performance on the original graph in the tables for easier comparison.
5. The discussion on limitations is insufficient. As the authors, you likely have a deep understanding of the potential weaknesses of your approach. A more thorough and reasonable analysis of the limitations would help readers better assess the scope and applicability of your work.

---

> ### Author Rebuttal · Authors · 2025-07-30
>
> We sincerely thank you for the detailed comments and insightful questions! Responses are as follows.
>
> ---
>
> **W1:** In Figure 1.C, consider showing how all three metrics change on the original graph before and after the attack, instead of only on the condensed graph, to better assess GC’s performance under attack.
>
> **A1**: Thank you for your insightful review.
> We present below the results showing how all three metrics change on the original (including non-condensed) graph, both before and after adversarial attacks. (*) indicates under attack.
>
> |  | Original Graph | Original Graph(*) | Condensed Graph | Condensed Graph(*) |
> | --- | --- | --- | --- | --- |
> | Intrinsic Dimension | 12.14 | 13.61 | 2.24 | 8.57 |
> | Fisher’s Discriminant Ratio | 0.19 | 0.22 | 0.09 | 0.41 |
> | Fraction of Hyperspheres Covering Data | 0.93 | 0.93 | 0.09 | 0.99 |
>
> As shown, adversarial attacks increase the classification complexity not only in the condensed graph but also in the original graph, especially in Intrinsic Dimension & Fisher’s Discriminant Ratio.
>
> ---
>
> **W2:** Section 3.2 aims to produce clearer decision boundaries via Curvature-Aware Manifold Smoothing. Given this, is the method in Section 3.3 necessary? The potential redundancy between the two remains unclear.
>
> **A2**: Thank you for your insightful review.
>
> - **Motivation:** Our framework is designed based on the observation that adversarial attacks increase the **classification complexity** of the condensed graph. According to classification complexity theory [1,2], this complexity arises from three main factors: **(1) higher intrinsic dimensionality, (2) increased class overlap, and (3) complex class boundaries.** Motivated by this, we introduced three targeted modules in our framework to reduce intrinsic dimension, class overlap, and boundary complexity respectively.
> - **Principle**: Curvature-aware smoothing is a **local** second‑order regularizer, it penalizes Gaussian curvature to make each class boundary smoother, **but it does *not* change where or how much different class manifolds overlap in space.** By contrast, class‑wise decoupling is a **global** volume‑based loss that directly minimizes the shared support between classes.
> - **Example:** Imagine class A’s points lie on a unit circle and class B’s on a concentric circle of radius 1.2, creating an annular overlap. Curvature smoothing can round off any zigzag in the decision curve, yet it still must traverse the overlapping band. Only the decoupling module, which effectively shrinks the inner ring or expands the outer ring, can open a gap so that a smooth boundary can pass between the two without crossing.
> - **Empirical:** We conducted an ablation study to evaluate the necessity of each component. Below are results comparing the full model (MRGC) with variants that omit either the curvature (w/o Curvature) or decoupling (w/o Decoupling) modules:
>
>
>     |  | **Cora(1.30%)** |  | **CiteSeer(0.90%)** |  |
>     | --- | --- | --- | --- | --- |
>     |  | Acc | Gap | Acc | Gap |
>     | MRGC | 77.43 | / | 65.12 | / |
>     | MRGC(w/o Curvature) | 76.01 | -1.42 | 64.57 | -0.55 |
>     | MRGC(w/o Decoupling) | 75.77 | -1.66 | 64.78 | -0.34 |
>
>     These results confirm that **both modules contribute meaningfully**, with curvature-aware smoothing providing more gain on **Cora** and decoupling having a greater impact on **CiteSeer**.
>
>
> *[1] How complex is your classification problem? a survey on measuring classification complexity. CSUR’19*
>
> *[2] Complexity measures of supervised classification problems. TPAMI’02*
>
> ---
>
> **W3:** A more precise or formal explanation of the manifold would clarify the intuition and theoretical basis.
>
> **A3**: Thank you for your insightful review.
>
> We clarify its definition as follows:
>
> $\mathcal{M}(G\')$ denotes the underlying data manifold on which the condensed graph $G\'$ is assumed to lie. This assumption is consistent with standard manifold learning literature[1][2], which suggests that high-dimensional data often reside on a lower-dimensional data manifold. In our context, we consider $\mathcal{M}(G\')$ as a smooth, compact submanifold embedded in $\mathbb{R}^{d'}$, formed by the learned node representations $\mathbf{Z}\' = (\mathbf{A}\')^2 \mathbf{X}\'$.
>
> *[1] Intrinsic dimension of data representations in deep neural networks, Neurips’19*
>
> *[2] Low dimensional manifold model for image processing, SIAM Journal on Imaging Sciences’17*
>
> ---
>
> **W4:** In Tables 1 and 2, including performance without condensation would help assess GC’s effectiveness under different attacks.
>
> **A4**: Thank you for your insightful review.
>
> The results corresponding to Tables 1 (Performance comparison under different condensation ratios when the training graph is corrupted) **without graph condensation** are provided below:
>
> | Dataset | Ratio | GCond | w/o Condensation | **MRGC** |
> | --- | --- | --- | --- | --- |
> | Cora (1,10,20) | 1.30% | 70.69 | 67.25 | **77.43** |
> | CiteSeer (1,10,20) | 0.90% | 60.77 | 59.94 | **65.12** |
> | PubMed (0.1,10,20) | 0.08% | 70.81 | 72.76 | **74.85** |
> | DBLP (0.1,10,20) | 0.11% | 62.50 | 57.74 | **65.11** |
> | Arxiv (0.1,10,20) | 0.05% | 58.16 | **65.86** | 59.38 |
>
> The results corresponding to Tables 2 (Performance comparison under different attack budgets when the training graph is corrupted) **without graph condensation** are provided below:
>
> | Dataset | Budget | GCond | w/o Condensation | **MRGC** |
> | --- | --- | --- | --- | --- |
> | Cora(1.30%) | S.5% | 60.59 | 59.54 | **65.57** |
> |  | S.10% | 59.09 | 50.89 | **64.02** |
> |  | F.20% | 69.36 | 68.31 | **74.79** |
> |  | F.30% | 65.22 | 65.48 | **72.36** |
> |  | L.30% | 64.97 | 62.80 | **70.47** |
> |  | L.40% | 57.49 | 47.95 | **63.89** |
> | CiteSeer(0.90%) | S.5% | 48.51 | 42.67 | **52.91** |
> |  | S.10% | 45.06 | 39.78 | **52.72** |
> |  | F.20% | 58.28 | 59.25 | **62.76** |
> |  | F.30% | 54.45 | 58.46 | **62.41** |
> |  | L.30% | 51.49 | 43.80 | **59.33** |
> |  | L.40% | 49.25 | 44.78 | **59.20** |
> | PubMed(0.08%) | S.0.5% | 56.54 | 56.74 | **60.57** |
> |  | S.1.0% | 55.21 | 53.70 | **58.96** |
> |  | F.20% | 71.10 | 67.81 | **73.43** |
> |  | F.30% | 67.26 | 62.83 | **68.86** |
> |  | L.30% | 68.04 | 57.59 | **73.38** |
> |  | L.40% | 56.14 | 37.45 | **58.28** |
> | DBLP(0.11%) | S.0.5% | 62.31 | 59.55 | **65.36** |
> |  | S.1.0% | 61.26 | 57.73 | **64.49** |
> |  | F.20% | 60.41 | 60.91 | **63.42** |
> |  | F.30% | 60.45 | 60.13 | **63.68** |
> |  | L.30% | 70.37 | 65.45 | **73.39** |
> |  | L.40% | 68.30 | 68.62 | **73.12** |
>
> As observed, vanilla graph condensation sometimes offers minor improvements to robustness under adversarial attacks, but can also fail to do so in certain cases. **In contrast, our MRGC consistently enhances robustness across most datasets and attack scenarios** (except on Arxiv).
>
> ---
>
> **W5 & L1:** The limitations discussion is considered insufficient.
>
> **A5**: Thank you for your insightful review.
>
> Here, we further discuss our limitations sufficiently: First, we have not yet evaluated the robustness of our method on graph classification tasks. Second, we have not tested its effectiveness under node injection attacks, where the number of nodes may vary.

---

### Official Review · Reviewer_gMV8 · 2025-07-02

**Clarity:** 3
**Significance:** 3
**Originality:** 3
**Rating:** 5
**Confidence:** 4

**Summary:**

This paper introduces Manifold-constrained Robust Graph Condensation (MRGC), a novel framework designed to enhance the robustness of Graph Condensation (GC) against universal adversarial attacks on graph features, structures, and labels. The authors empirically and theoretically demonstrate that GC inherently reduces the classification complexity of graphs, a critical property that is highly vulnerable to adversarial perturbations. To counter this vulnerability, MRGC incorporates three graph data manifold learning modules: Intrinsic Dimension Manifold Regularization, Curvature-Aware Manifold Smoothing, and Class-Wise Manifold Decoupling. These modules guide the condensed graph to reside within a smooth, low-dimensional manifold with minimal class ambiguity, thereby preserving GC's classification complexity reduction capability and ensuring robust performance against diverse attack scenarios. Extensive experiments show that MRGC consistently outperforms existing baselines across various datasets and attack budgets

**Questions:**

- The experiments evaluate robustness against specific and commonly used poisoning attacks: PRBCD for structure, random feature sampling, and uniform label flipping. I am not confidence on the attack performance of random feature sampling and uniform label flipping. I am curious will the resilience still hold after being attacked by more dedicated attack methods like like Nettack[1].

[1] Zügner, Daniel, Amir Akbarnejad, and Stephan Günnemann. "Adversarial attacks on neural networks for graph data." Proceedings of the 24th ACM SIGKDD international conference on knowledge discovery & data mining. 2018.

**Ethical Concerns:**

["NO or VERY MINOR ethics concerns only"]

**Limitations:**

**Scope of Adversarial Attacks:** The experiments evaluate robustness against specific and commonly used poisoning attacks: PRBCD for structure, random feature sampling, and uniform label flipping. The study does not investigate the method's resilience against a broader range of more sophisticated or adaptive adversarial attacks.

**Hyperparameter Tuning:** The MRGC framework introduces three new regularization hyperparameters (α,β,γ) which must be tuned. The paper notes that these were determined through a grid search, which can be a computationally intensive process and may pose a practical challenge for applying the method to new datasets.

**Quality:**

3

**Strengths And Weaknesses:**

**Strength:**
- Novelty in Robustness: The paper addresses a largely unexplored area: the robustness of Graph Condensation (GC) against various adversarial attacks (feature, structure, and label perturbations).

- Clear presentation of the designed modules.

- Comprehensive analysis and experimentation: It provides both empirical investigation and theoretical analysis to reveal that GC is an intrinsic-dimension-reducing process, and this property is vulnerable to adversarial attacks.

- Demonstrated Superior Performance: Extensive experiments show that MRGC consistently outperforms state-of-the-art GC methods and robust graph denoising techniques across various datasets and attack budgets.

- It is appreciated that the author conducts the ablation study on the MRGC performance by removing each module.

---

> ### Author Rebuttal · Authors · 2025-07-30
>
> We sincerely thank you for the detailed comments and insightful questions! Responses are as follows.
>
> ---
>
> **W1 & L1: It's unclear whether MRGC hold under attacks like Nettack.**
>
> **A1**: Thank you for your insightful review.
>
> We agree that robustness under more adversarial attacks is crucial for assessing the reliability of our MRGC. However, Nettack is a **targeted attack** designed to perturb specific individual nodesin the original graph. Since graph condensation produces a compact synthetic graph that does not maintain one-to-one correspondence with original nodes, targeted attacks like Nettack are not directly work in this setting.
> To address the core of your insightful concern, we further evaluate the robustness of **MRGC** under **MetaAttack** [1], **a global adversarial attack**. The following table presents the test accuracy of GNNs trained on condensed graphs generated by different condensation methods under MetaAttack perturbations:
>
> |  | **Cora(1.30%)** | **CiteSeer(0.90%)** | **PubMed(0.80%)** |
> | --- | --- | --- | --- |
> | GCond | 73.53 | 57.77 | 50.87 |
> | SGDD | 71.27 | 48.62 | 41.63 |
> | GEOM | 45.8 | 35.86 | 44.12 |
> | RobGC | 74.14 | 58.54 | 49.43 |
> | **MRGC** | **77.17(+3.03%)** | **64.23(+5.69%)** | **52.67(+2.19%)** |
>
> As shown in the table, **MRGC consistently achieves the best robustness across all datasets under MetaAttack**, outperforming both standard and robustness-aware baselines.
>
> *[1] Adversarial Attacks on Graph Neural Networks via Meta Learning. ICLR’19*
>
> ---
>
> **L2: Tuning three hyperparameters via grid search adds computational overhead.**
>
> **A2**:  Thank you for your insightful review.
>
> While three hyperparameters need to be tuned, **most optimal values remain consistent across datasets**, reducing the need for exhaustive search. Empirically, we also find that **smaller values are generally effective when the condensed graph contains fewer nodes per class**, providing practical guidance to streamline tuning in new settings.

---

> > ### Comment · Reviewer_gMV8 · 2025-08-09
> >
> > Thank the authors for addressing my questions, I am good to keep my score as accept!

---

### Comment · Area_Chair_aPtf · 2025-08-07
**Reminder: reviewer–author discussion phase is approaching its end**

Dear Reviewers,

This is a friendly reminder that the reviewer–author discussion phase is approaching its end. If you have not yet responded to the author's rebuttal, please do so as soon as possible. If your questions have been resolved, kindly acknowledge that in the discussion thread. If not, please indicate what remains unclear.

Before submitting the “Mandatory Acknowledgement,” please ensure that you have engaged in actual communication with the authors. Thank you again for your time and efforts.

Regards,

AC

---

### Note · Authors · 2025-08-12

We sincerely appreciate the time and effort from all the reviewers and the ACs in providing us with insightful and constructive feedback!

**All reviewers have consistently expressed supportive opinions about our work and believe that our paper merits clear acceptance**. They are in agreement regarding our contributions and strengths, which can be summarized as follows:

1. **First robust condensation framework:** We are the first to investigate the robustness of graph condensation, which is an underexplored, important problem. [`#gMV8, #GwSS, #EHXr`]
2. **Novel insight and perspective:** We are the first to connect and explore the relationship between intrinsic dimension (ID) and condensation, offering a new perspective to the field.[`#gMV8, #GwSS, #EHXr, #6kqs`]
3. **Superior Performance**: Our proposed MRGC consistently outperforms state-of-the-art GC methods and robust graph denoising techniques across various datasets and attack budgets.[`#gMV8, #GwSS, #EHXr`]
4. **Clear presentation:** Our writing clearly conveys the core motivation and contributions of the work.[`#gMV8, #GwSS`]

While a few minor concerns were raised (regarding robustness against additional adversarial attacks, the individual contributions of our three proposed modules, and the more detailed explanation of our theorem), **we have addressed all of their concerns thoroughly in the rebuttal stage.**

**All reviewers have acknowledged our clarifications during the rebuttal, recognized the contributions of our work, and believed our work can be clearly accepted.**

Once again, we sincerely thank all the reviewers and the AC for their time and effort!

---

### Decision · Program_Chairs · 2025-09-17

**Decision:**

Accept (spotlight)

**Comment:**

The paper introduces a manifold-constrained framework for robust graph condensation that addresses the vulnerability of condensed graphs to adversarial perturbations. All reviewers agree that the work addresses an important and underexplored problem, which offers novel theoretical insights, and extensive experiments demonstrate robustness gains. Initial concerns include the scope of attacks, clarity of the theory, necessity of individual modules, possible trade-offs on clean graphs, etc. The rebuttal provides additional experiments, clearer definitions, and in-depth justifications. Reviewers acknowledge these clarifications in their follow-up. Overall, it is concluded that the paper is technically solid, clearly presented, and makes a valuable contribution.